# Deep learning to decode sites of RNA translation in normal and cancerous tissues

Jim Clauwaert [1,2,3] ✉, Zahra McVey[4], Ramneek Gupta [4], Ian Yannuzzi[5], Venkatesha Basrur[6], Alexey I. Nesvizhskii[6,7], Gerben Menschaert [8,9] ✉ & John R. Prensner [1,2,3,9] ✉

The biological process of RNA translation is fundamental to cellular life and has wide-ranging implications for human disease. Accurate delineation of RNA translation variation represents a significant challenge due to the complexity of the process and technical limitations. Here, we introduce RiboTIE, a transformer model-based approach designed to enhance the analysis of ribosome profiling data. Unlike existing methods, RiboTIE leverages raw ribosome profiling counts directly to robustly detect translated open reading frames (ORFs) with high precision and sensitivity, evaluated on a diverse set of datasets. We demonstrate that RiboTIE successfully recapitulates known findings and provides novel insights into the regulation of RNA translation in both normal brain and medulloblastoma cancer samples. Our results suggest that RiboTIE is a versatile tool that can significantly improve the accuracy and depth of Ribo-Seq data analysis, thereby advancing our understanding of protein synthesis and its implications in disease.

RNA translation is an intricate process initiated by the stepwise binding of the 40S and 60S ribosome subunits to RNA, along with multiple eukaryotic initiation factors and other cofactors, to form an actively translating ribosome[1]. RNA translation is a major determinant of protein abundance and represents a core area of disease biology including cancer, where numerous genetic and non-genetic factors alter the composition of ribosomes, efficiency of translation, and fidelity of translation[2].

To gain global insights into ribosome activity, ribosome profiling (Ribo-Seq) has become increasingly popular to determine the translational efficiency of mRNAs and detect non-canonical open reading frames (ORFs) and alternative proteoforms that have eluded standard analyses[3,4]. The computational analysis of Ribo-Seq data has therefore become a cornerstone of research fields that rely on accurate identification of translated ORFs for various purposes, including de novo gene discovery, RNA regulation, proteogenomics, microprotein

biology, and disease-focused research on therapeutic agents whose mechanism targets RNA translation. Yet, Ribo-Seq data analyses have been challenged by a lack of statistical power for detecting small ORFs and variable patterns between mapped ribosome read patterns and translation, a phenomenon caused by both biological (e.g., tissue type, cell lines vs. tissue samples) and technical (e.g., translation inhibitors, lab protocols) factors of the experiment[3]. This variability poses challenges for statistical tests that apply manually curated features to derive the presence of translated ORFs from the data, such as those used by existing tools (Supplementary Tables 1–4; e.g., ORFquant[5], Rp-Bp[6], Ribo-TISH[7], Ribotricer[8], PRICE[9] and RibORF[10]). Consequently, evaluation of existing tools reveals significant disagreement and an abundance of false positive predictions[11,12].

Here, we have created RiboTIE (Fig. 1a) to improve the analysis of translated ORFs using Ribo-Seq data. RiboTIE applies state-of-the-art techniques and best practices in machine learning to improve the

[1]Division of Pediatric Hematology/Oncology, Department of Pediatrics, University of Michigan, Ann Arbor, MI, USA. [2]Chad Carr Pediatric Brain Tumor Center, University of Michigan, Ann Arbor, MI, USA. [3]Department of Biological Chemistry, University of Michigan, Ann Arbor, MI, USA. [4]Novo Nordisk Research Centre Oxford, Novo Nordisk Ltd, Oxford, UK. [5]Cancer Program, Broad Institute of MIT and Harvard, Cambridge, MA, USA. [6]Department of Pathology, University of Michigan, Ann Arbor, MI, USA. [7]Department of Computational Medicine and Bioinformatics, University of Michigan, Ann Arbor, MI, USA. [8]Department of Data Analysis and Mathematical Modelling, Ghent University, Ghent, Belgium. [9]These authors contributed equally: Gerben Menschaert, John R. Prensner. ✉e-mail: clauwaer@umich.edu; gerben.menschaert@ugent.be; prensner@umich.edu

detection of translated ORF delineation through Ribo-Seq data. Our tool is designed specifically to process large biological omics data, can leverage pre-trained models, and does not suffer from lack of power for detecting small translated ORFs. RiboTIE is able to capture correlations between ribosome protected fragments (RPFs) and translated ORFs that are specific to individual datasets, which allows robust performance across diverse Ribo-Seq datasets. Application of RiboTIE to human cancer samples and cell lines from patients with medulloblastoma recapitulates known disease findings and reveals robust tool performance across sample quality characteristics. We propose that RiboTIE offers a versatile tool for the analysis of Ribo-Seq data for biologists and bioinformaticians seeking to expand insights into human cell and disease states.

## Results

RiboTIE is characterized by several unique aspects: first, RiboTIE omits the requirement of several common pre-processing steps, such as the determination of read length offsets between the 5'-read-end and A-site of RPFs (Supplementary Figs. 1–5), which we evaluated to also lack a gold standard. Thus, instead of adjusting or discarding any mapped reads based on the overall alignment properties of different read lengths, RiboTIE processes all reads by position and length, resulting in an overall performance improvement (Supplementary Fig. 6).

Second, in contrast to other tools[5–10], RiboTIE constructs ORFs after the prediction step rather than before it. Specifically, RiboTIE processes ribosome information along the full transcriptome and predicts the presence of translation initiation sites (TISs) for each possible codon in any given RNA molecule. By evaluating every position as a candidate TIS, all possible ORFs on the transcriptome are scored, which facilitates downstream processing and benchmarking of the tool. Selection of ORFs adhering to certain properties (e.g., an AUG start codon) are thus selected as a last step. This approach also increases the training set as it is able to apply to the full transcriptome.

Third, RiboTIE only processes the counts of mapped RPFs along the transcript and therefore does not consider related sequence information (e.g., start codon) or ORF characteristics (e.g., length), which may propagate potential biases for determining translated ORFs.

Fourth, RiboTIE applies recent advances in machine learning (transformer networks) that are tailored to processing variable-length inputs (i.e., mapped read counts along the transcript) and automated feature extraction. Furthermore, these approaches have enabled us to use pre-trained models optimized on numerous ribosome profiling samples, which we observe increases overall performance (Supplementary Table 5).

To assess RiboTIE, we benchmarked our tool on eight Ribo-Seq experiments derived using varying treatment methods (Supplementary Dataset 1). Here, we compared RiboTIE against multiple common tools for translated ORF detection[5–10]. To do this, we evaluate RiboTIE on the full set of predictions generated by each of the other tools, with each tool having different implementations libraries created. Indeed, RiboTIE provides predictions for all possible ORFs, where a one-on-one comparison between RiboTIE and the full ORF library of each tool allows for a viable benchmarking strategy. It is important to note that, since each ORF library is made out of a different positive and negative set, listed performances between any of the previous tools cannot be compared. Evaluating the ability of each tool to correctly identify canonical CDSs (positive set) in each ORF library, we found RiboTIE to be more sensitive and precise, as reflected by the Area Under the Receiver Operating Characteristics (ROC AUC) and Precision Recall (PR AUC) curves (Fig. 1b; Supplementary Dataset 2; Supplementary Fig. 7).

To further analyze the strengths and weaknesses of each tool in addition to evaluating the PR/ROC AUC scores, we compared their positive predictions in more detail using six biological replicates of pancreatic progenitor cells (Supplementary Fig. 8)[13]. We found that our tool retrieved 64.9% more CDSs (31,431) as compared ORFquant (19,064), which obtains the second most calls for annotated CDSs (Fig. 1c). For smaller CDSs less than 300 bp in length, RiboTIE retrieved 300% more CDSs (4043) as compared to ORFquant (1001). RiboTIE reproducibly called 50% of annotated CDSs for each of the six datasets, where only an average of 4.2% of the calls are unique within each of the replicate samples (Fig. 1c). We found that RiboTIE retrieved the largest quantity of annotated CDSs with non-canonical start sites across all six datasets (48; all CUG), where ORFquant (1; AUA) and PRICE (21; all CUG) are the only other tools that also feature non-canonical start codons (see "*Methods*" for details on tool parameters). We also found notable differences between the number and types of nominated non-canonical ORFs (Fig. 1d; Supplementary Figs. 9–12; Supplementary Dataset 3). Notably, RiboTIE has a substantially higher fraction of upstream (overlapping) ORFs (u(o)ORFs). The fraction of internal ORFs (intORFs) and downstream (overlapping) ORFs (d(o)ORFs) is low with RiboTIE, in contrast to Ribo-TISH and Ribotricer, as prediction of these ORFs is known to be plagued by false-positive calls[3,11]. Additionally, RiboTIE calls considerably fewer lncRNA-ORFs compared to other high-performing tools such as ORFquant and Rp-Bp. Nonetheless, for these three tools, the called lncRNA-ORFs follow similar distributions: ~25% of called lncRNA-ORFs have TISs that overlap with protein-coding sequences, whereas ~46% overlap with exons from protein-coding transcripts (Supplementary Fig. 13). Evaluation of the reads-per-base counts (as a metric of ORF abundance level) and model output distributions of RiboTIE revealed that lncRNA-ORFs with lower read counts had lower RiboTIE model scores (Supplementary Fig. 14), where lncRNA-ORFs called by other tools fall under the threshold of 0.15 applied for RiboTIE throughout this study.

We next sought to apply RiboTIE to human tissue samples, where data quality may be more variable compared to cell line experiments. We evaluated 73 brain samples from both fetal (30) and adult (43) patients[14] along with 15 medulloblastoma patient tissues[15] (Fig. 2a). Notably, data quality is poor for some of the samples, with total in-frame read occupancies below 60% in 32 samples and below 50% in 12 samples (Fig. 2a). Across the 73 fetal and adult normal brain samples, RiboTIE made a total of 63,786 unique ORF calls (Supplementary Fig. 8b), of which 29,988 (47.0%) were annotated CDSs and 9407 ncORFs (14.7%) (Fig. 2b). This represented a substantial performance improvement relative to a much larger number of called ORFs (158,855, of which 28.9% are annotated CDSs and 30.8% ncORFs) previously reported for the same dataset through the RibORF software[14].

Across these data, RiboTIE calls 36 CDSs with a non-AUG start codon as compared to 9 such instances by RibORF. For the ncORFs, the calls of RiboTIE were largely dominated by AUG start codons, whereas RibORF returns mostly non-canonical start codons (Fig. 2c). Interestingly, for the 73 evaluated brain samples that have varying in-frame read occupancies between 36% and 75% (Fig. 2a), we find only a slight and non-significant correlation with RiboTIE's performance (Pearson $r = 0.227$; $p = 0.054$) (Fig. 2d). As a strong correlation (Pearson $r = 0.804$; $p = 1.1e - 17$) does exist between the number of mapped reads within coding sequences and RiboTIE's performance (Fig. 2e), our results indicate that reads that appear out-of-frame for technical and sample processing or quality reasons, are equally leveraged by RiboTIE to determine translated ORFs. In addition, there is an even stronger correlation when only considering the number of reads around the TISs (±30 nt) and RiboTIE's performance (Pearson $r = 0.903$; $p = 7.5e - 28$).

Following this observation, we generated ribosome profiling data from medulloblastoma cell lines either treated with dimethyl sulfoxide (DMSO) or homoharringtonine (HHT). HHT blocks translation and concentrates RPFs around TISs. Compared to DMSO, we find an increased number of CDSs retrieved for 12 out of 15 pairs of

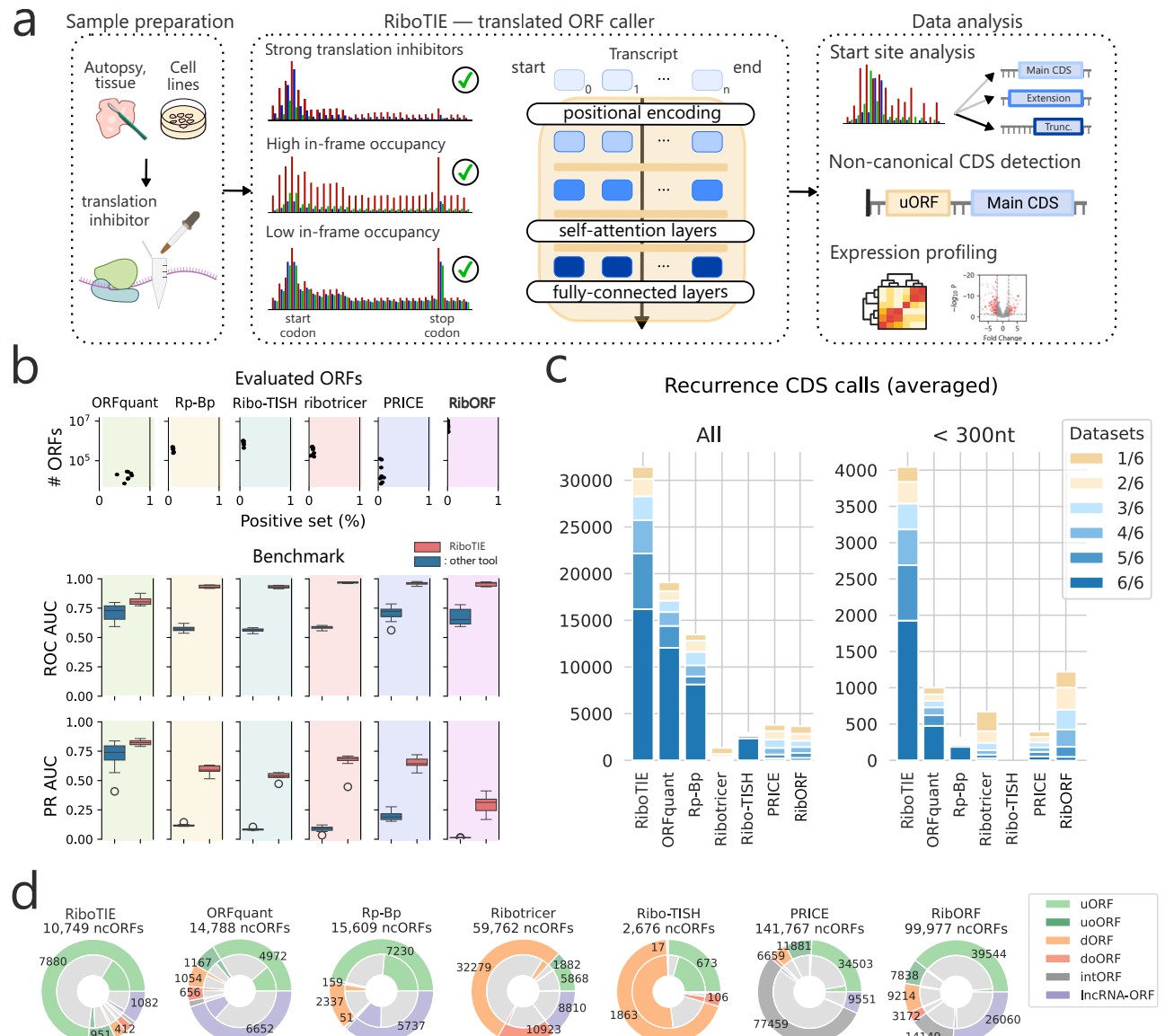

**Fig. 1 | RiboTIE combines flexibility and performance for the detection of RNA translation in Ribo-Seq data. a** Schematic outlining RiboTIE's function for the analysis of translated products (CDS coding sequence, uORF upstream open reading frame). (left) RiboTIE has been tested on Ribo-Seq from various cell types and translation inhibitors, where (middle) transformer models processing read counts aligned to transcripts are capable of leveraging information from both data with low and high in-frame read occupancy. (right) RiboTIE can be applied for a variety of studies, including start site analysis, detection of ncORFs, and expression profiling. **b** Benchmarking analyses featuring eight datasets. RiboTIE is compared with six other tools for translated ORF delineation from ribosome profiling.

Precision recall (PR) or Receiver Operator Characteristic (ROC) Area Under the Curve (AUC) scores are compared on ORF libraries that are generated by and unique to each tool. Data in box plots show the median (line), 25th to 75th percentiles (box), and 1.5 times interquartile range (whiskers). **c** A stacked barplot that reflects the number of called annotated CDSs (left, all; right, <300 nt) for each ORF caller tool for six replicate samples of pancreatic progenitor cells, the fraction of CDSs that are found in a certain number of replicates is represented as well. **d** The total number of non-canonical ORFs (ncORFs) and each type of ncORFs called by each tool combining all predictions on the six replicate samples of pancreatic progenitor cells. The inner fractions represent ncORFs present in >4 datasets.

HHT-treated cell lines (Fig. 2f). Including 15 medulloblastoma tissue samples from a previous study[15], we find HHT-treated cells to improve performance when incorporating the effect of read depth between samples (ANCOVA $p = 4.05e - 11$; DF = 1) (Fig. 2g).

To illustrate the role of ncORFs in medulloblastoma, we processed a total of 24 medulloblastoma cell line samples utilizing RiboTIE to evaluate differentially expressed ncORFs between samples with high ($n = 16$) and low ($n = 8$) *MYC* expression, which is used to classify distinct medulloblastoma subtypes[16]. Across all datasets, a total of 3,638 ncORFs, of which 69.4% are upstream ORFs (uORFs), were selected for evaluation. We found 201 ncORFs with substantial alterations ($|\text{Fold Change}| > 2$; $p_{\text{adj}} < 0.05$) in translational expression (Fig. 2h;

Supplementary Dataset 4). Notably, these ncORFs resided on transcripts which also reflected established medulloblastoma biology, with enrichment for genes with known tissue-enriched expression in the nervous system and differentiation-related genes such as *NEUROD1* implicated in medulloblastoma (Supplementary Fig. 15a–c)[17]. We further employed published RNAseq data for 39 medulloblastoma patient tissues[18] to characterize the differential translational abundance of RiboTIE-identified ncORFs with differential RNA expression of their mRNA transcripts according to *MYC* status (Supplementary Fig. 15d). Therefore, RiboTIE nominations may provide direct links to disease biology.

To provide a more focused inspection of top ncORF candidates in medulloblastoma, we integrated RiboTIE-identified ORFs with ORF

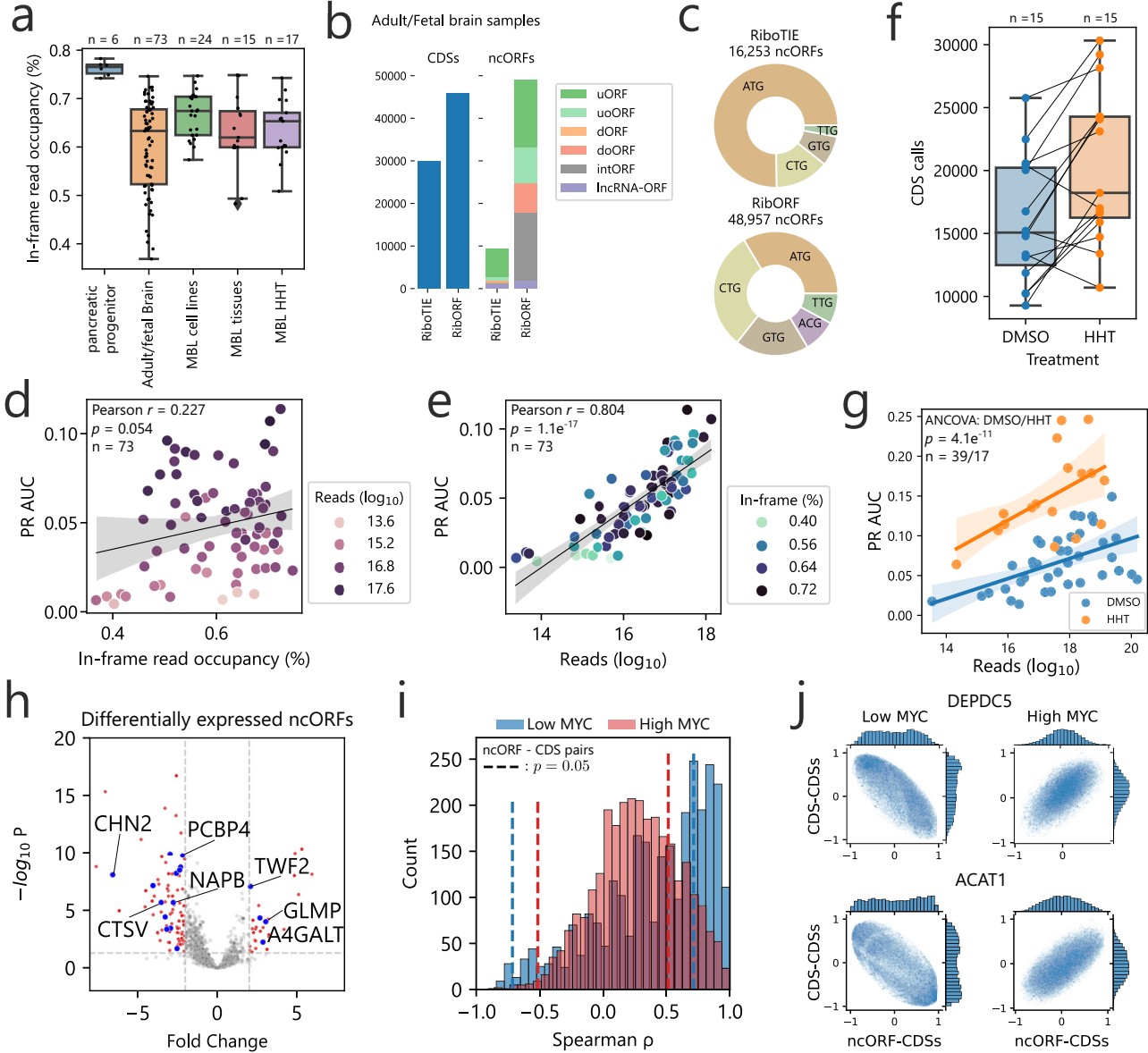

**Fig. 2 | Application of RiboTIE to human normal tissues and brain cancer for improved analysis of RNA translation. a** Box plot showing the in-frame read occupancy (reads mapped to reading-frame by 5'-end vs. total reads within CDSs) for all data applied in this study (See Supplementary Dataset 1; MBL medulloblastoma). **b** Bar plot displaying the combined number of unique calls for annotated coding sequences (CDSs) and non-canonical ORFs (ncORFs) on 73 adult/fetal brain samples as reported by the original paper[14] (RibORF) and RiboTIE. **c** A pie chart on the start codon distribution of all called ncORFs for the 73 adult/fetal brain samples. **d** Scatter plot displaying the Pearson $r$ statistic and fitted linear regression function between the Area Under the Precision Recall curve (PR AUC) of RiboTIE on adult/fetal brain samples ($n = 73$) and mapped reads on the transcriptome and **e** in-frame read occupancy. **f** Number of CDSs called by RiboTIE outlined by both a scatter plot and box plot for medulloblastoma cell lines ($n = 15$) treated with Dimethylsulfoxide (DMSO) control or homoharringtonine (HHT). Identical cell lines are linked. **g** Scatter and fitted linear regression plot on 39 DMSO (blue) and 17 HHT (orange) medulloblastoma samples shows the improved performance of

RiboTIE on HHT-treated cells. P-value by ANCOVA analysis between DMSO and HHT, with mapped reads as covariate. **h** Volcano plot showing differential expression of called ncORFs of low MYC ($n = 8$) as compared to high MYC ($n = 15$) expressing medulloblastoma cell lines. Threshold lines denote $p_{adj} < 0.05$ (y-axis) and |Fold Change|>2 (x-axis). Blue dots accompanied by listed gene names are ncORFs confirmed by TIS Transformer. Differential expression analysis was performed with a two-sided Wald test. $P$ values were adjusted for multiple comparisons using the Benjamini-Hochberg method. **i** Histogram showing ρ correlations of a two-sided Spearman test existent between ncORFs and their matching CDSs for both low MYC (blue) and high MYC (red) cell lines. Threshold lines denote $p = 0.05$. **j** Scatter plots of two-sided Spearman rank correlations between the ncORF or downstream CDS and all other CDSs on the genome for both low and high MYC expression (*DEPDC5/ACAT1*). All error bands given denote a 95% confidence interval. All box plots show the median (line), 25th–75th percentiles (box), and data within 1.5 times interquartile range (whiskers).

predictions by TIS Transformer, our prior machine learning tool that detects ORFs with a high chance of translation based on the RNA sequence context[1]. The use of TIS Transformer alongside RiboTIE offers verification of candidate ORFs based both on sequence context and measured ribosome footprints. We found a total of 22 candidate ncORFs supported by both RiboTIE and TIS Transformer with

differential translation between medulloblastoma subtypes (Fig. 2h, annotated with blue dots). In accordance with known disease biology, translational abundance of these ncORFs correctly clustered medulloblastoma cell lines by their disease subtype (Supplementary Fig. 16). None of the selected ncORFs reflected lncRNA-ORFs, and are thus positioned close to, or overlapping with, annotated CDSs.

Overall, the majority of ncORF-CDS pairs were positively correlated (Fig. 2i). We observed 38 ncORF-CDSs pairs that showed a negative correlation (Spearman $\rho < -0.71; p < 0.05$) for low *MYC* expression and 25 pairs for high *MYC* expression (Spearman $\rho < -0.50; p < 0.05$) (Supplementary Dataset 4). These ncORF-CDS pairs with discordant correlations were enriched for genes associated with known protein acetylation (Supplementary Fig. 15e; Supplementary Dataset 5). Interestingly, we found 9 ncORF-CDS pairs in 7 genes that are inversely correlated between cell lines with low and high *MYC* expression (all $p < 0.05$), where three are validated on sequence context using TIS Transformer (*DEPDC5, ACAT1, MPHOSPH6*; Fig. 2j). To probe this further, we observed that 4 of these 7 genes demonstrated post-transcriptional regulation in medulloblastoma patient tumors, in which *MYC*-high tumors demonstrated discordant CDS regulation in proteomic data compared to RNAseq data (Supplementary Fig. 16b, c). We conclude that application of RiboTIE in medulloblastoma enables biological insight into gene regulation observed in human disease.

Lastly, we sought to generate orthogonal evidence for RiboTIE-nominated ncORFs. We generated deep tryptic mass spectrometry data for three *MYC*-high and three *MYC*-low medulloblastoma cell lines and queried for peptides to ncORFs using a subgroup-specific FDR of 1%. We found a total of 44 peptides supporting ncORFs or unannotated N-terminal extensions of CDSs, which was similar to the number of peptides detected for ORFquant-nominated events (Supplementary Dataset 6 and Supplementary Fig. 17). We found MS to primarily return matches on N-terminal extensions when applied on ORF called by RiboTIE as compared to lncRNA-ORFs when assessed against ORFquant calls (Supplementary Fig. 17), in line with abundances observed on previous datasets (Supplementary Fig. 8). Intriguingly, we observed several peptides that associated with *MYC*-high compared to *MYC*-low status (Fig. 3). Among them was a peptide supporting an intORF nested within the *SCRIB* gene CDS, even though the *SCRIB* CDS was not differentially abundant (Fig. 3a). We also observed proteoforms such as a N-terminal extension of the *RBMS1* CDS, which was downregulated in MYC-high samples along with the annotated CDS (Fig. 3b), and an uoORF in the *ZNF717* gene, which was not significantly changed in abundance in *MYC*-high vs *MYC*-low samples even though the *ZNF717* CDS was (Fig. 3c). Overall, these results highlight the potential for ncORFs to exhibit differential disease patterns in medulloblastoma compared to their associated canonical CDS.

## Discussion

RiboTIE is a machine learning tool that detects ORFs from ribosome profiling fingerprints that combines the newest advances and adheres closely to "best practices" in the field. The tool is designed to be precise, robust, modular, and transparent in its function and has shown to outperform existing tools substantially. Importantly, RiboTIE does not pre-process the data towards P-sites of ribosomes which was shown to improve performance. We believe offsetting mapped read positions per read length could introduce biases or reject valuable information as offsets from the 5'-end are shown to be non-deterministic for a given read length (Supplementary Fig. 2), where RiboTIE performs well on data with low in-frame read percentages when evaluating the 5'-ends of each read. While this pre-processing step may improve accuracy for tools employing manually curated features, we observe that automated feature extraction with machine learning is able to overcome sole dependence on in-frame read percentages. Lastly, we find that the distribution of detected ORF types remains stable across samples with differing sequencing depths (Supplementary Fig. 18), where the number of total called translated ORFs is mainly affected. This showcases RiboTIE's ability to return stable accuracies on the called translated ORFs (i.e., fraction of annotated CDS's discovered), even in settings with low coverage.

Importantly, RiboTIE determines translated TISs from which ORFs are derived as a post-processing step. While RiboTIE performs better as compared to other tools (Fig. 1b, c), we were not able to validate whether this approach adversely affected the model's ability to resolve transcript isoforms resulting from internal exon splicing. In these cases, incorporation of RNA-seq data or evaluation of the uniformity of the ribosome reads across the full ORF may be necessary. While machine learning becomes more widespread as a means to construct performant tools, the dependence on GPU hardware remains the main limitation for making RiboTIE widely accessible. An additional challenge on the subject of translated ORF calling, and a future focus on the development of the tool, is the inclusion of an additional post-processing approach for grouping and resolving ORF calls on transcripts that share genomic regions with a high number of other transcript isoforms to flag likely false positives (e.g., by exclusion of transcripts tagged as nonsense mediated decays or retained intron; Supplementary Fig. 9) and optimizing prediction thresholds for ORF types with low read counts (Supplementary Fig. 14). Notably, any improvements to the post-processing algorithm won't require the need to re-run the fine-tuning or predictive step of RiboTIE, showcasing the flexible design of the tool.

To conclude, we have widely applied RiboTIE across 166 datasets with varying sequencing depths ($2.7E + 5 - 2.5E + 8$ reads; Supplementary Dataset 1), demonstrating its unique ability to handle data with low in-frame read occupancy and demonstrating its ability to refine biological insights for disease subtypes of childhood medulloblastoma. RiboTIE offers a new avenue to spearhead studies on translation start site analysis, non-canonical ORF detection and expression profiling of translated ORFs using Ribo-Seq data. RiboTIE is available as a Python package (see "*Code Availability*") with pre-trained models that allow fast optimization times on new data.

## Methods

### Medulloblastoma cell culture

In our study, we utilized the following cell lines with their respective sources and catalog numbers: DAOY (ATCC, Cat#HTB-186), CHLA-01-MEDR (ATCC, Cat#CRL-3034), D425 (Bandopadhayay lab), D458 (Bandopadhayay lab), D341 (ATCC, Cat#HTB-187), D384 (CCLE), MB002 (Bandopadhayay lab), R256 (CCLE), R262 (CCLE), CHLA-259 (Children's Oncology Group, Cat#CHLA-259), Med2112 (Brain Tumor Resource Lab), Med411 (Brain Tumor Resource Lab), D283Med (ATCC, Cat#HTB-185), ONS76 (CCLE), and UW228 (CCLE). Cell lines were routinely verified via STR genotyping and tested for mycoplasma contamination using the Lonza MycoAlert assay (Lonza) CHLA-01-MEDR, Med2112 (expressing mCherry and luciferase), Med411 (expressing GFP and luciferase), and MB002 cells were maintained in Tumor Stem Media comprised of DMEM/F12 (1:1) with Neurobasal-A medium (Invitrogen) and supplemented with HEPES (1 M, 0.1% final concentration), sodium pyruvate (1 mM final concentration), MEM non-essential amino acids (0.1 mM final concentration), GlutaMax (1 × final concentration), B27 supplement (1 × final concentration), human EGF (20 ng/mL), human FGF-basic-152 (20 ng/mL), and heparin solution (2 μg/mL final concentration). D283, D341, D384, D425, D458, DAOY, R262, UW228 and CHLA-259 cells were maintained in DMEM supplemented with 10% FBS and 1% penicillin-streptomycin in a 5% $CO_2$ cell culture incubator. ONS76 cells were maintained in RPMI 1640 supplemented with 10% FBS and 1% penicillin-streptomycin in a 5% $CO_2$ cell culture incubator.

### Medulloblastoma ribosome profiling data

Published medulloblastoma cell line (DMSO treated) and tissue data were obtained from the Short Read Archive (PRJNA957428, for cell lines) or dbGAP (phs003446, for tissues). Matched homoharringtonine ribosome profiling data was generated for D283, DAOY, ONS76, R262,

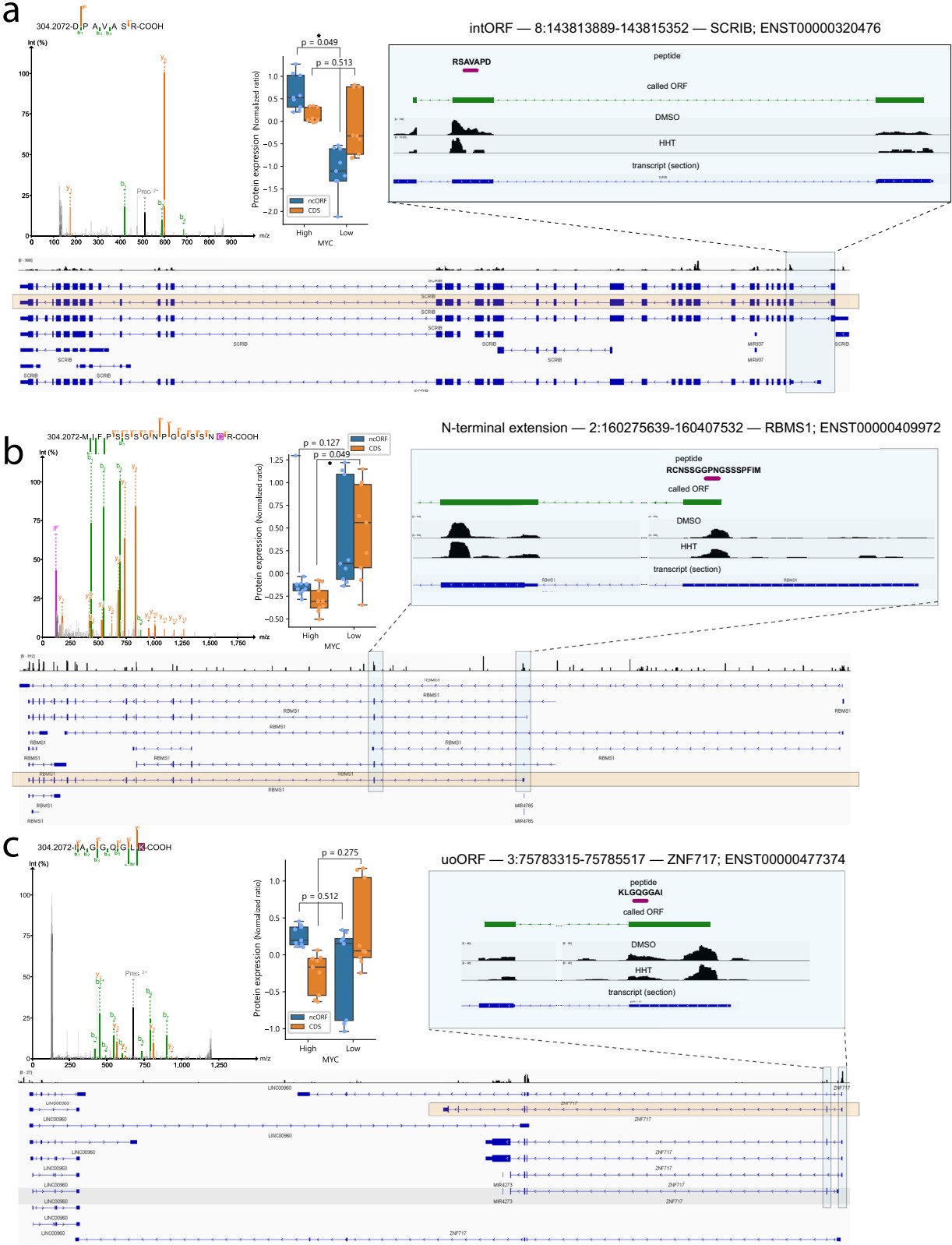

**Fig. 3 | Example ncORFs called by RiboTIE for which MS evidence is found.** Mass spectrometry is performed on 6 medulloblastoma cell lines ($n = 3$ MYC-high and $n = 3$ MYC-low) in technical triplicate, for 18 total data points (Supplementary Dataset 6). **a** an intORF in the SCRIB gene in shown. **b** an N-terminal extension of the RBMS1 gene is shown. **c** an uoORF in the ZNF717 gene is shown. For the three called ORF examples, each image shows (left) the mass spectrometry spectrum, (middle) the protein abundance values of the coding sequence (CDS) and non-canonical open reading frame (ncORF; e.g., intORF internal ORF, uoORF upstream overlapping ORF) for MYC-high and MYC-low cell lines, and (right; bottom), the location of the peptide and the called ncORF in relation to the canonical coding sequence. P values are calculated by a two-sided Mann Whitney $U$ test comparing the averaged values of MYC-high ($n = 3$) vs MYC-low ($n = 3$) cell lines. All box plots show the median (line), 25th–75th percentiles (box), and data within 1.5 times interquartile range (whiskers).

UW228, D384, D458, D425, Med411, SUMB002, Med2112, CHLA-01-MEDR, and D341. Ribosome profiling was performed as detailed previously[15]. In brief, cells were grown to 60–70% confluence. Cells were then pre-treated with 5 μg/mL homoharringtonine for 10 min. Cells were then harvested, washed once in 1x cold PBS, and collected by centrifugation. Cells were lysed in a buffer of 20 mM Tris HCl, 150 mM NaCl, 5 mM MgCl2, 1 mM dithiotrietol, 0.05% NP-40 and 25 U/mL Turbo-DNase I (Invitrogen). 2 μg/mL of cyclohexamide was additionally added to the lysis buffer. Cleared lysates were quantified and 2.5U/μg of RNAse I was added for 45 min at room temperature for digestion. The reaction was quenched with an equal volume of 1U/μL Superase RNase Inhibitor (Ambion). RPFs were isolated using ultracentrifugation at 259,000 × g at 4C for 2 h. RPF RNA was purified with a Zymo Direct-Zol kit (Zymo) and rRNA was depleted using the siTOOLS human riboPOOL kit (siTOOLS Biotech, Germany). Denatured RPFs were resolved on a 15% TBE-Urea gel (200 V, 65 min) and the RPF band at 26–32 nt was excised and RNA extracted. After RNA precipitation, RNA was end-repaired with T4 PNK at 37 °C for 1 h, cleaned up with RNA Clean and Concentrator kit (Zymo). The 3' linker adapter was ligated (22 °C, 3 h) using T4 RNA ligase I and R4 RNA ligase 2 deletion mutant. Linker reactions were removed with Rec J exonuclease (Lucigen) and 5' deadenylase (NEB). cDNA was synthesized with EpiScript RT enzyme (Lucigen, 50 °C for 30 min). After clean-up with Exonuclease I and RNAse I and hybridase (Lucigen), cDNA was resolved on a 10% TBE-Urea gel (175 V for 1 h), and 75nt bands were excised and DNA was precipitated. Precipitated cDNA was circularized with CircLigase (Lucigen, 60 °C for 3 h). Circular cDNA was amplified using 2x Phusion HiFi master mix (NEB) in a 11–14 cycle PCR reaction depending on cDNA yield. PCR products were precipitated with ethanol and 5 M NaCl, and resolved on a 8% TBE gel (100 V, 90 min). The 150 nt band was excised, purified, quantified for DNA abundance using a DNA Qubit, and analyzed for library size using an Agilent TapeStation (Agilent). Libraries were then sequenced on an Illumina NovaSeq.

## Ribosome profiling pre-processing

All generated and publicly available data (Supplementary Dataset 1) were mapped and processed using an identical approach. Ensembl assembly GRCh38 version 110 was used as the reference transcriptome. Trimming was performed using cutadapt version 4.4 filtering out reads smaller than 14nt (-m 14)[19]. Next, STAR version 2.7.11a[20] was used to filter contaminant RNA and DNA using the special arguments "SeedSearchStartLmaxOverLread .5", "outFilterMultimapNmax 1000", "outFilterMismatchNmax 2", and "outReadsUnmapped Fastx", and to map the left out reads to the transcriptome using the flags "quantMode TranscriptomeSAM", "outFilterMultimapNmax 10", "outMultimapperOrder Random", "outFilterMismatchNmax 2", and "seedSearchStartLmaxOverLread 0.5", and "alignEndsType EndToEnd".

For further processing with RiboTIE, the aligned read files are read where the number of mapped reads per read length and position on the transcripts is used to create the vector embeddings for the model (Supplementary Figs. 6, 19). For ORFquant, Rp-Bp, Ribo-TISH, Ribotricer, PRICE, and RibORF, the recommended default settings are applied (see Supplementary Methods).

## Vector embeddings from mapped reads

RiboTIE processes mapped RPF counts along the full transcript in parallel. No pre-processing of other types of data, such as start codon or ORF information, are used to curate features or build a candidate ORF library. Instead, only the mapped position (5'-end) and length of mapped RPFs is utilized (Supplementary Figs. 3–5). To allow computation with a transformer-based architecture, vector representations are calculated that represent this information for each position on the transcriptome. For a given position, the vector embedding $e_c$ is obtained from the read count $c$ and a set of feed-forward layers $\phi$. Existing tools use a similar methodology but apply various approaches

to offset mapped reads as a function of their read length (Supplementary Table 2). Read counts are normalized across the transcript for numerical stability.

$$e_c = e \odot \tanh(\phi(c)) \tag{1}$$

With $c \in [0, 1]$, $\phi : R^1 \rightarrow R^h$, and $e \in R^h$. $h$ is a hyperparameter of the model indicating the input dimension. In this paper, we explore the inclusion of ribosome read length information as part of the information applied to determine TIS locations. For a given transcript position, $e_l$ is calculated using the read length fractions $l$ between 20 and 41 following the equation:

$$e_l = \sum_{i=0}^{21} E_i * l_i \tag{2}$$

with $E \in R^{21 \times h}$ and $l \in [0, 1]^{21}$, where $\sum l = 1$. The matrix $E$ incorporates vector embeddings for read lengths 20–40 and is optimized as part of the training process. Note that ribosome data by read length is sparse and the majority of values in $l$ are 0. After evaluation of different input embeddings (Supplementary Table 3), we find the optimal vector embedding for the model at each position to be $e_c + e_l$.

## Model architecture and optimization

RiboTIE is created to map the translatome at single nucleotide resolution of samples using ribosome profiling data. To simplify the experimental learning objective of detecting translated ORF regions, we train a model to detect active TISs, denoting a binary classification problem. ORFs are afterwards derived using a greedy selection of the first in-frame stop codon.

Continuing upon our previous work using transformer networks for the detection of TISs using transcript sequence information[1], we similarly applied the Performer architecture for its memory efficiency which allows calculation of the attention matrix spanning full transcript sequences[21].

To cover the full transcriptome without overfitting the machine learning model on the target labels, two models are trained on non-overlapping folds of the transcriptome. The training, validation and test sets are constructed from transcripts grouped per chromosome to ensure all transcript isoforms are grouped together. Models are trained on the training set (chromosomes [3, 5, 7, 11, 13, 15, 19, 21, X], [2, 6, 8, 10, 14, 16, 18, 22, Y]) until a minimum loss on the validation ([1, 9, 17][4, 12, 20]), set is reached, where results are acquired from the test set ([2, 4, 6, 8, 10, 12, 14, 16, 18, 20, 22, Y, KL…,], [1, 3, 5, 7, 9, 11, 13, 15, 17, 19, 21, X]). This approach, in which two models are optimized with the test sets covering the full transcriptome through non-overlapping folds is followed for all datasets in the paper. The outputs of both models are combined to attain a full set of predictions on the complete transcriptome. Allocations as defined here are applied for all reported results in the manuscript.

Following the use of Ensemble GRCh38 to map the ribosome profiling data, we have derived both the transcriptome regions and positive set from the Ensembl assembly. This totals ~250 k transcripts and ~431 M transcript nucleotide positions on which each sample is mapped, where annotated TISs ("start codon") positions are utilized as the positive set when optimizing and evaluating the model.

Hyperparameter tuning was performed through evaluation of standard hyperparameters (e.g., size of the hidden dimension, number of layers, number of attention heads per layer), evaluating numerous architectures ranging from 81 K to 525 K model parameters. No individual hyperparameters were observed to be more effective than others in improving performances. Rather, a correlation exists between the total number of model parameters and model performance, where we found the optimal architecture to feature 212 K weights (Supplementary Methods, Supplementary Table 4, Supplementary Fig. 20).

## Model pre-training and fine-tuning

Following up on the success of using pre-trained models in the field of natural language processing, we evaluated two pre-training schemes in this study. Specifically, models are first optimized on a single dataset derived from a multitude of ribosome profiling samples (dubbed the pre-training step). Afterwards, the optimized weights are used to instantiate training on individual data sets to correctly capture correlations of the ribosome profiling signal that is unique to each sample (dubbed the fine-tuning step). Identical data allocations of the chromosomes as outlined in the previous section were applied for both the pre-training and fine-tuning step following the use of GRCh38.

The self-supervised training objective optimizes a model to infer the presence of mapped reads at a given position and read length. Hence, labels are derived from the ribosome profiling data. The positive labels are allocated when more than one read of that length is mapped to that position, resulting in a binary classification task. During the training process, 15% of input positions are randomly selected and masked (Supplementary Fig. 21). For the supervised pre-training, an identical optimization scheme is used as the fine-tuning step, optimizing the model directly to predict the presence of TISs.

Eight datasets were applied for the pre-training objective and eight for the fine-tuning step for the benchmark. All data were selected to cover a variety of tissues and treatment methods (Supplementary Dataset 1), where the data is processed using Ensembl (GRCh38). Pre-training approaches improved performances and sped up convergence times as compared to training models from scratch, where the supervised training objective showed to be the most effective (Supplementary Table 4, Supplementary Figs. 22, 23). All results in the main manuscript have been derived from fine-tuning on these evaluated supervised pre-trained models.

## Benchmarking with existing translated ORF callers

Existing tools that call translated ORFs from Ribo-Seq data each evaluate custom sets of ORFs (i.e., ORF libraries) that are derived from both the mapped ribosome reads and reference assembly. In this manuscript, we compare RiboTIE against ORFquant, Rp-Bp, Ribo-TISH, Ribotricer, PRICE, and RibORF. For all results, the Ensembl assembly (GRCh38) is used to map the reads and derive the positive/negative sets.

## Sensitivity and precision on full ORF library

RiboTIE returns predictions for the full transcriptome, which allows evaluation of the performance of RiboTIE against each tool using its complete ORF library constructed for each sample. ROC/PR AUC values were calculated for each of eight benchmark datasets to include various dataset conditions (Fig. 1b; Supplementary Dataset 1; Supplementary Fig. 7). Due to the different markup of ORF libraries between tools, where larger sets with fewer positive samples typically reflect a more difficult evaluation set, ROC/PR AUC performances across tools are not representative of their respective abilities to detect translated ORFs. For each tool, the recommended settings are used (Supplementary Methods). To compare the output of the statistical method itself without any post-processing steps, the full set of ORFs was evaluated for Rp-Bp (--write-unfiltered) and RibORF (pred.*), as opposed to the reduced set of ORFs determined from overlapping ORFs. While other tools, such as ORFquant, are also likely to include a filtering step at the end, other tools did not give access to these results. Importantly, the reduced set of Rp-Bp and RiboTIE are used for any of the other analyses in this paper.

## Pancreatic progenitor cells, fetal and adult brain tissue cells, and medulloblastoma tissue samples

By default, RiboTIE performs some additional steps to ensure the quality of proposed translated ORFs. First, a positive set is constructed from predictions with a score above 0.15. ORFs not adhering to a valid start codon (not *TG; ~10% of unfiltered set with score >0.15) and those without a stop codon on transcript (~5% of unfiltered set with score >0.15; excludes many transcript isoforms labeled as nonsense-mediated decay etc.) are excluded from the final set. It is observed that, for transcripts featuring fewer mapped reads around the translation initiation site, RiboTIE is more prone to miss TISs by a multiple of three bases (Supplementary Fig. 24). To address predictions on these samples, a neighborhood searching step is performed when creating the result table that corrects non-ATG predictions to in-frame ATG positions if present within a 9 codons distance (~15% of set with score >0.15). This filtering was not performed to obtain the ROC/PR curves (last paragraph) but have been applied for the pancreatic progenitor cells, fetal and adult brain tissue samples, and medulloblastoma samples (Supplementary Dataset 1).

For the pancreatic progenitor cells, the top scoring ORFs for each tool were evaluated. Importantly, tools vary by providing either predictions on a smaller positive set (e.g., ORFquant) or the complete evaluated ORF library (e.g., ribotricer). For tools returning predictions for a very large set of ORFs, a positive set of ORFs is selected according to p-value statistic with the aim to create a set similar in size to the set of ORFs provided by RiboTIE (i.e., ribotricer, Ribo-TISH, RibORF). The reduced set of ORFs provided by Rp-Bp (.filtered) and RibORF (repre.*) were used for this analysis.

For the results on the 73 adult/fetal brain samples, the predicted set provided by Duffy et al.[14] was used to represent predictions for RibORF. To mirror the approach followed by Duffy et al. the positive set of called ORFs only includes ORFs that have been called on more than one dataset using RiboTIE.

"ncORF" are defined as either u(o)ORFs, d(o)ORFs, intORFs or lncRNA-ORFs. "lncRNA-ORFs" are assigned to ORFs on transcripts with the long non-coding RNA transcript tag as provided by GRCh38, with the added condition that they do not share their genomic start or stop site with any annotated CDSs. ORFs that share their genomic start or stop site with an annotated CDS and do not share the transcript with any canonical CDS are classified as "CDS variant". To ensure differences in called ORF types between tools are not a result of naming conventions, all predictions provided by each tool have been run through our annotation pipelines.

## Mapping differential expression and correlations between ncORF-CDS pairs

From a total of 26,437 ncORFs predicted by RiboTIE on all 24 medulloblastoma datasets, a subset of 5436 ncORFs is obtained by selecting predictions that are present in at least 5 experiments. Combining this set with all canonical CDSs, differential expression is performed using the default workflow offered by PyDESeq2 0.4.4, where the number of mapped reads between samples that feature low (8) and high (16) expression of the MYC gene are compared. Differentially translated ncORFs were selected considering significance and the degree of change ($|$Fold Change$|$>2; $p_{adj}$ < 0.05). A further filtered down list of ncORFs is given by applying a non-strict condition based on the output of TIS transformer (model output > 0.01).

Correlations were calculated on the set of 5436 ncORFs and their CDS counterparts. Here, the $\rho$ coefficients and p-values of a two-sided Spearman test are calculated from mapped ribosome reads using the Transcripts Per Million (TPM) normalization method. A further filtered down list of anti-correlated ncORFs-CDS pairs is derived using TIS transformer (model output > 0.01).

## Cross evaluation of in-house CRISPR screen on ncORFs in medulloblastoma cell lines

For analysis on the medulloblastoma cell line samples, we cross-evaluated a CRISPR screen that was previously designed for an in-house study on medulloblastoma cell lines. Results were evaluated for

RiboTIE and ORFquant by looking at the amino acid sequence overlap between the ncORFs screened and those called in this study[15].

## Gene network analysis

The 190 differentially-abundant ncORFs were considered for analysis (Supplementary Fig. 15). Network analysis was performed using the String-db tool[22] according to default settings using the total human proteome as a reference comparator.

## Medulloblastoma tissue analysis

Individual ncORFs with differential translation (Supplementary Fig. 15) were mapped to their respective Ensembl gene identifiers and queried using a matched set of mass spectometry and RNAseq for 39 medulloblastoma patient tissues from the Clinical Proteomic Tumor Analysis Consortium[18]. Log2-normalized data were stratified according to patients with the Group 3a medulloblastoma subtype, which were identified as the set with high MYC expression, and all other patients. Protein and RNA dyssynchrony was assessed by subtracting proteome data from RNAseq data values, to get a normalized score representing the discordance between the two data types. Correlation of ncORF-harboring mRNAs with MYC expression was performed using RNAseq data with a Spearman correlation.

## Mass spectrometry on medulloblastoma cell lines

**Protein digestion and TMT labeling.** MYC-high (MB002, D458, D283,) and MYC-low (ONS76, DAOY, UW228) cells were grown as described. 20 million cells each were washed with cold PBS twice and lysed with RIPA buffer, quantified with a bicinchoninic acid (BCA) assay, adjusted to 2µg/µL and separated into technical triplicates. Samples were submitted to Proteomics Resource Facility at the University of Michigan for processing and mass spectrometry data acquisition. Briefly, upon reduction (5 mM DTT, for 30 min at 45 °C) and alkylation (15 mM 2-chloroacetamide, for 30 min at room temperature) of cysteines in samples, the proteins (50 µg/condition) were precipitated by adding 6 volumes of ice-cold acetone followed by overnight incubation at −20 °C. The precipitate was spun down, and the pellet was resuspended in 0.1 M TEAB and overnight (-16 h) digestion with trypsin/Lys-C mix (1:40 protease:protein; Promega) at 37 °C was performed with constant mixing using a thermomixer. The TMT 18-plex reagents were dissolved in 20 µl of anhydrous acetonitrile and labeling was performed by transferring the entire digest to TMT reagent vial and incubating at room temperature for 1 h. Reaction was quenched by adding 5 µl of 5% hydroxyl amine and further 15 min incubation. Labeled samples were mixed and dried using a vacufuge. An offline fractionation of the combined sample (-300 µg) into 24 fractions was performed using high pH reversed-phase chromatography (Zorbax 300Extend-C18, 2.1 mm × 150 mm column on an Agilent 1260 Infinity II HPLC system). Fractions were dried and reconstituted in 9 µl of 0.1% formic acid/2% acetonitrile in preparation for LC-MS/MS analysis. TMT channel labeling was the following: D458 rep 1 (126), D458 rep 2 (127N), D458 rep 3 (128N), D283 rep 1 (129N), D283 rep 2 (130N), D283 rep 3 (131N), MB002 rep 1 (132N), MB002 rep 2 (133N), MB002 rep 3 (134N), DAOY rep 1 (135N), DAOY rep 2 (127C), DAOY rep 3 (128C), ONS76 rep 1 (129C), ONS76 rep 2 (130C), ONS76 rep 3 (131C), UW228 rep 1 (132C), UW228 rep 2 (133C), UW228 (134C).

**Liquid chromatography-mass spectrometry analysis.** Orbitrap Ascend Tribrid mass spectrometer equipped with FAIMS source and Vanquish Neo UHPLC was used to acquire the data (Thermo Fisher Scientific). Two µl of the sample was resolved on an Easy-Spray Pep-Map Neo column (75 µm i.d. × 50 cm; Thermo Scientific) at the flow-rate of 300 nl/min using 0.1% formic acid/acetonitrile gradient system (3–19% acetonitrile in 72 min;19–29% acetonitrile in 28 min; 29–41% in 20 min followed by 10 min column wash at 95% acetonitrile and

re-equilibration) and directly spray onto the mass spectrometer using EasySpray source (Thermo Fisher Scientific). FAIMS source was operated in standard resolution mode, with a nitrogen gas flow of 4.2 L/min, and inner and outer electrode temperature of 100 °C and dispersion voltage or −5000 V. Three compensation voltages (CVs) of −40, −55 and −70 V, 1 s per CV, were employed to select ions that enter the mass spectrometer for MS1 scan and MS/MS cycles. Mass spectrometer was set to collect MS1 scan (Orbitrap; 400–1600 m/z; 120 K resolution; AGC target of 100%; max IT in Auto) following which precursor ions with charge states of 2–6 were isolated by quadrupole mass filter at 0.7 m/z width and fragmented by High-energy C-trap dissociation (45 K resolution; collision energy 38%; normalized AGC target of 200%; max IT 50 ms).

**FragPipe data processing.** The data processing was performed using the FragPipe software using the recommended default settings[23], the Swiss-Prot protein database employed by Fragpipe was enriched independently with the protein sequences derived from ncORFs called by both RiboTIE and ORFquant. Peptides derived from called ncORFs were given a lower confidence score (PE = 10) in order to exclude peptide matches that also align with proteins listed in Swiss-Prot (PE ∈ [1,5]). In addition, all proteins derived from ncORFs for which resulting peptides were equal to or a substring of any canonical proteins of the Ensembl assembly were filtered out to ensure that matched peptides align to unique findings.

## Reporting summary

Further information on research design is available in the Nature Portfolio Reporting Summary linked to this article.

## Data availability

The Medulloblastoma HHT Ribo-seq data generated in this study have been deposited in the Gene Expression Omnibus repository under accession code PRJNA1077309. The Mass spectrometry data generated in this study has been deposited in the ProteomeXchange under accession code PXD055854. Other existing data applied throughout the study includes those for the benchmark (SRR1802129, SRR2433794, SRR2732970, SRR2733100, SRR2954800, SRR8449577, SRR9113067, SRR11005875), model pre-training (SRR592960, SRR1562539, SRR1573939, SRR1610244, SRR1976443, SRR2536856, SRR2873532, SRR3575904), pancreatic progenitor cells (GSE144682), medulloblastoma cell lines (PRJNA957428) and tissue samples (phs003446), and fetal/adult brain samples (phs002489).

## Code availability

RiboTIE (https://doi.org/10.5281/zenodo.10689717)[24] is implemented in Python and is available through GitHub (https://github.com/TRISTAN-ORF/RiboTIE) and PyPI (https://pypi.org/project/transcript-transformer/).

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

## Acknowledgements

We thank Pratiti Bandopadhayay and Joelle Straehla for sharing medulloblastoma cell lines. We thank Sebastiaan van Heesch for helpful comments on the manuscript. J.R.P. acknowledges funding from the National Institutes of Health/National Cancer Institute (K08-CA263552-01A1), the Alex's Lemonade Stand Foundation Young Investigator Award (#21-23983), the St. Baldrick's Foundation Scholar Award (#931638), the DIPG/DMG Research Funding Alliance, the Book for Hope Foundation, the Yuvaan Tiwari Foundation, the Hyundai Hope on Wheels Foundation, the ChadTough Defeat DIPG Foundation, the Andrew McDonough B+ Foundation, the Curing Kids Cancer Foundation, and a Collaborative Pediatric Cancer Research Awards Program/Kids Join the Fight award (#22FN23). G.M. acknowledges support from Novo Nordisk. J.R.P. is the Ben and Catherine Ivy Foundation Clinical Investigator of the Damon Runyon Cancer Research Foundation [CI-127-24].

## Author contributions

Conceptualization: J.C., J.R.P., G.M.; methodology, J.C., J.R.P., I.Y., G.M.; formal analysis, J.C.; investigation, J.C., Z.M., R.G., J.R.P., I.Y., V.B., A.I.N., G.M.; resources, J.R.P., G.M.; data curation, J.R.P., J.C.; writing—original draft, J.C., J.R.P., G.M.; writing—review & editing, J.C. and J.R.P. with input from all authors; visualization, J.C., J.R.P., G.M.; supervision, J.R.P., G.M.; project administration: J.C., J.R.P., G.M..; funding acquisition, J.R.P., G.M.

## Competing interests

G.M. is an employee of OHMX Bio. Z.M. and R.G. are employees of Novo Nordisk Ltd. J.R.P. reports receiving honoraria from Novartis Biosciences. J.R.P. is a paid consultant for ProFound Therapeutics. A.I.N. receives royalties from the University of Michigan for the sale of MSFragger and IonQuant software licenses to commercial entities. The remaining authors declare no competing interests.
