## [Transparent Peer Review file · Nature Communications]

Deep learning to decode sites of RNA translation in normal and cancerous tissues

Corresponding Author: Dr John Prensner

Version 0:

Reviewer comments:

Reviewer #1

(Remarks to the Author)

In this study, Clauwaert et al. present RiboTIE, a deep learning-based method to predict novel coding sequences using ribosome profiling data. The RiboTIE pipeline takes the mapped ribosome protected fragments (RPFs) as the input and uses a transformer model to predict the translation initiation sites (TISs) at single-nucleotide resolution. The model was pre-trained on a collection of human ribosome profiling datasets covering different tissues and experimental protocols. Then it was fine-tuned on the target dataset, which requires less computational resource and allows the model to learn dataset-specific rules. The authors demonstrated RiboTIE's performance for the detection of translated ORFs against a few other computational tools. In addition, they conducted ribosome profiling experiments on medulloblastoma cell lines and identified differentially translated non-canonical ORFs (ncORFs) between high/low MYC expression levels. It remains a challenge to decipher the full coding potential of the human genome, and sophisticated tools are needed to improve our understanding of the translational landscape in health and disease. The authors should address the following concerns before publication.

Major comments

1. One of the main concerns I have is regarding the advantages of RiboTIE over TIS transformer, developed by some authors of this work (PMID: 36879896). It seems that both approaches are designed to predict translation initiation sites, use the same transformer architecture called performer and have been trained on the same dataset -- TIS from the Ensembl dataset. On a similar note, in Fig. 2h, the authors used TIS transformer to "confirm" the ncORF predictions by RiboTIE. Does the usage of both tools increase the sensitivity or accuracy of the predictions? The authors should provide guidance on the best strategy to use RiboTIE.

2. Existing Ensembl annotations may suffer from limitations like under-representation of short and unstable proteins. I wonder how this potential bias affect the prediction of coding sequences. Are the newly predicted translation start sites more enriched in certain transcript biotypes or UTR regions compared to the training dataset? In addition, the authors have proposed some interesting ideas like pretraining the neural network first and then fine-tuned it on separate datasets (e.g., self-supervised and supervised pre-training). How does this affect novel ORF predictions aside from the difference in validation loss?

3. Is there a specific reason that RibORF is not included in the benchmarking analyses in Fig. 1b? Apparently, RibORF has been cited and used in this work (e.g., Fig. 2b and 2c).

4. The Precision recall (PR AUC) scores reported in Supplementary Table 3 and 4 (0.1-0.2) are significantly lower than the values in Fig. 1b (> 0.5). Are the authors using the same model for these benchmarking experiments? I understand that for each method in Fig. 1b, different ORF subsets were used to evaluate model performance, but how does this filtering step affect RiboTIE's performance?

5. Supplementary Figure 5 makes little sense to me. The fluctuations in the validation loss are expected due to the stochastic nature of neural network training. But the differences in the model performance seem to be comparable to noise. The authors should use approaches like cross-validation tests to produce replicable results.

6. It is unclear how the authors' findings in Fig. 2h and 2j could link back to known or novel cancer biology. I appreciate the authors' effort in conducting new cancer cell line experiments. However, the biological implications of the RiboTIE results

are not fully explored. What might be the functional consequences of these differentially translated ncORFs? This work would benefit from a more-in-depth analysis of these findings.

7. The language used in this paper is too generic. For example, the authors mentioned “Current computational approaches to Ribo-Seq have been developed on a central statistical test using manually curated features...” (p3) and “We have developed RiboTIE as a best-in-class and highly versatile analysis tool ...” (p5). It should be noted that there are numerous computational tools designed for very different tasks in ribosome profiling data analysis. The authors are encouraged to precisely describe the specific task that RiboTIE focused on.

8. I suggest that the authors rewrite the method section to include more details. For example, very little information is given for the training of three types of models (no-pretraining, supervised pre-training and self-supervised pretraining). How are the training datasets constructed in the pretraining and fine-tuning phases? Which model is used to generate results in Fig. 1 and Fig. 2.? In the supplementary material (p3, 1.3.3), the authors mentioned that “multiple models are trained”, so how many models are trained? Also, technical details are missing for Extended Data Figure 7. How are the PCA and t-SNE analysis done?

9. The authors should consider expanding the main text of this paper. For example, a brief introduction to the existing approaches to delineate ORFs based on ribosome profiling data and mRNA sequence would be beneficial. Additionally, the authors should discuss the potential limitations of their approach.

10. Multiple names are used for this work (https://github.com/jdcla/RIBO_former). The authors should use a consistent name for their approach.

Minor comments

1. In discussion part (p5), “highly versatile analysis tool approach...” seems redundant.
2. Figure 2h, tick labels in y axis should be positive.
3. Figure 2j legend, “(SNAPC5/ACAT1)” I believe it should be “(DEPDC5/ACAT1)”.
4. Supplementary Table 4: “Position” should be “Pre-train”.
5. X axis label is missing for Extended Data Figure 1a.
6. Colormap should be shown for Extended Data Figure 1b.
7. In Extended Data Figure 6, what are colored dots?
8. Extended Data Figure 7, maybe SSH is mistaken for SHH?

(Remarks on code availability)

Reviewer #2

(Remarks to the Author)

The authors present a transformer-based method to identify sites of translation using Ribo-seq data agnostic of start-codon information or ORF characteristics. This is a great approach and a transformer approach is a valuable addition to tools that are currently available for ORF calling. Particularly, powerful as the approach is able to recover ORFs without using the sequence information i.e., without making a restrictive ORF reference library. Although, since there have been several tools published previously, it would be useful to see orthogonal methods of validations of ORFs identified by RiboTIE such as MS, CRISPR-screens etc.. Lastly, overall, the manuscript is well written but the supplementary methods and figure legends are very brief and need to be expanded.

Major comments

- 1) While it is not necessary that all the ORFs make stable peptides, it would be important to show if the ORFs identified by RiboTIE are found more often as functional or stable peptides. Independent validations such as using either newly generated or published A) CRISPR-screen data and B) Mass-spectrometry/Immuno-peptidomics data to show detection rate of RiboTIE smORFs vs other tools.
- 2) Similarly to point 1, how evolutionary conserved are the identified smORFs and how often do they have other sequence features such as protein domains as compared to other methods?
- 3) According to Fig. 1A the tool also finds peptide extensions of annotated CDS. Are the peptide extensions of annotated CDSs found by RiboTIE also found in MS data, i.e., particularly the sequence of the peptide extended 5' of the annotated start?
- 4) How many start isoforms of the known CDS does each tool find? Is the discovery rate higher in general for RiboTIE? How many times did RiboTIE find a TIS for an annotated CDS that was outside of the annotated TIS?
- 5) Certain regions of the transcriptome are not mappable considering sequence constraint and the short length of Ribo-seq reads. How do such regions affect the model? Could the authors show how many times annotated ORFs with such unmappable regions are still called by RiboTIE in comparison to other tools?
- 6) Could the authors show some examples of some CDSs captured by their method but were not detected by other methods? Plotting a nucleotide resolution coverage view of such examples will help to understand the strengths of RiboTIE specifically in when it is able to detect smORFs that are missed by other tools. Similarly, it would help to show some examples where other tools find a smORF but RiboTIE misses detecting them.
- 7) On the github page it shows the authors have incorporated a 9-codon search to select alternative ATGs if a non-ATG is identified as a start. It was unclear if this was incorporated in the manuscript - would help to expand on the methods section

explaining this. In relation to the same question, how do the authors deal with alternative-start isoforms? Are there multiple isoforms with a different start but having the same splice or same stop codon considered a different ORF?

8) Apart from showing start-codon distribution of RiboTIE identified smORFs, could the authors also show other properties of the smORFs identified such as length distribution, expression and so on.

9) In order to understand what the model is capturing, it would be useful to test and highlight what are the main features the model is capturing such as coverage, length of ORF, read length distribution etc.?

10) Is one of the main features of the model ORF expression? In that case, is this the reason for low detectability of ncORFs as these are generally found to have lower expression levels compared to uORFs? It would be useful to highlight this as a limitation if that is the case.

11) It is unclear for Fig 1b as to how the ROC curve is constructed? Is it that, for each tool's ROC curve, various true sets based on other tools are considered as a positive set? Could the authors draw a Supplementary figure with these ROC curves as well as elaborate in the methods or Supplementary figure legend as to how these were derived to make it easier to understand.

12) In general, the methods section in main/supplementary is quite brief and needs to be expanded.

13) Many of the figure legends do not fully explain the details of the figures. For Fig. 1A it would help to explain briefly for what's in each box in the caption. For Extended data figure 1b, what is meant by coverage in the y axis is unclear. For extended data figure 3a it has not been explained what ec, el, E0 etc., mean? For extended data figure 4, please split into A and B and describe accordingly as these are referenced in different text sections. For extended data figure 6 what do the coloured dots mean?

14) Could the authors please elaborate on why the TIS transformer model was combined with the RiboTIE approach - what are the limitations of RiboTIE approach that needed both these approaches to be incorporated to find the final set of smORFs for the medulloblastoma dataset? On intersection of the two methods, why did it lead to a large reduction of numbers?

Corrections

15) In Fig.1b it seems that RiboTISH should have had ~33% positive set as per supp data file but in the figure it looks close to < 10%. Please check and rectify.

16) Also, in the supp file 2: The Rb-bp has positive set 60-90% by calculation but it seems should have been <10%. Please check and rectify.

17) Extended data figure 4 - For the adult brain dataset the numbers don't match the main text. In the figure is it really union or intersect? Please check and rectify.

18) In Fig 2j, the legend says snapc5 gene but in the figure it says depdc5? Are these same genes or is this a typo - please check and rectify.

Minor comments

19) By the statement "We found that RiboTIE retrieved 64.9% more CDSs (31,431)...", do the authors mean start-sites? It was unclear whether the authors are comparing start identification or full ORF identification considering only one start-isoform. Please clarify in the text.

20) The title says detection of sites of RNA translation but the tool does not cover frameshifts or read throughs. It is focused on start-site identification and corresponding ORFs. It would be helpful to be clearer through the title, abstract and introduction that this is a TIS identifier/ ORF caller tool.

21) In Fig 1b, do the authors mean y axis to be the number of ORFs and not "samples"? It might be confusing to readers as is because it sounds as if the plot shows positive set ORFs found for X number of samples.

22) The authors use uORF/ncORF interchangeably - would help to be consistent. Especially when referring to ncORF-CDS pairs do the authors mean uORF or is it including lncORF with neighbouring protein-coding CDS pairs?

(Remarks on code availability)

The code is well documented on Github and I am able to install it. But there is an error on running the test set given using the current version, so I was unable to evaluate the tool.

Reviewer #3

(Remarks to the Author)

The ribo-seq technique has become indispensable for detecting the transcriptome and its variations, and more particularly for discovering translation events in unconventional ORFs. Several computational methods already exist for the analyses of Ribo-seq data. In this manuscript, the authors create RiboTIE, a method based on transformer models for mapping the global translation of RNA. Their findings indicate that RiboTIE provides higher accuracy and sensitivity in the analysis of ribosome profiling data, compared to other tools. They apply RiboTIE to ribo-seq analyses of normal brain and medulloblastoma cancer samples.

The creative use of the transformer architecture applied to RNA sequences and small ribosome-protected fragments is exactly the sort of cross-field transfer that promises to augment the analytical power needed to tackle the fast-accumulating amounts of data generated by recent advances in omics biosciences. We appreciate the extensive description of the algorithm's development methodology including the different schemes attempted for data representation, hyperparameter selection, and pre-training optimization strategies.

This new machine learning-based method will likely set a new standard in ribo-seq analysis.

The following comments are minor:

1. Introduction, second paragraph: the authors allude to certain challenges for computational analyses of Ribo-seq data. However, the provided information is very generic and the non-specialists in this field cannot understand what these challenges are. We therefore recommend specifying these challenges to understand the rationale of this work better.
2. First sentence: "RNA translation is an intricate process that involves the stepwise binding of the 40S and 60S ribosome subunits to RNA, along with multiple eukaryotic initiation factors and other cofactors". This sentence describes part of the translation initiation process, not the actual RNA translation process. Please modify accordingly.
3. We appreciate that both ROC AUC and PR AUC were provided as performance indicators. It may perhaps be beneficial to indicate that PR is a more telling measure given the strong imbalance between the classes if the audience is anticipated to comprise many non-experts in machine learning.
4. Fig 2b: why wasn't RibORF included in the benchmarking analyses? Indeed, the authors subsequently (fig 2b,c) compared some features of translated ORFs detected with RiboTIE against those detected with RibORF using the same dataset (reference 14).
5. Fig 2d, 2e and 2g: the indicated log10 values appear to be unusually high. Please check.
6. Supplementary figure 2, SRR1802129, graphs 36 and 37 length: problem with the labels, Y axis.
Supplementary figure 3, SRR8449577, graph 38 length: problem with the labels, Y axis.
Supplementary figure 4, SRR11005875, graphs 36 and 38 length: problem with the labels, Y axis.
7. Data availability: PRJNA1077309 is not available.

(Remarks on code availability)

Version 1:

Reviewer comments:

Reviewer #1

(Remarks to the Author)

I appreciate the authors' revisions to the manuscript, which have addressed most of my concerns. However, I still have a few comments regarding the current form of the manuscript before it can be published.

First, in their response to my fourth question, the authors mentioned: "The question of why Supplementary Tables 3-4 and Fig. 1b exhibit somewhat different PR AUC scores relates to the different sequencing depths of the different datasets, and thus comparing across datasets is not informative."

As a potential user, I kindly disagree with this statement, since I am interested in understanding when and how I can apply a tool like RiboTIE to my dataset. Sequencing depth is likely to be a key determining factor.

Second, the authors' response to my fifth question is not fully convincing. I have some expertise in the field of deep learning, and I still do not see why architecture 4 is the optimal. Furthermore, the authors stated: "The proper way to evaluate whether the differences in model performances are due to noise would be to repeat the training and evaluation phase, which we did many times as part of the overall process of conducting this study." If that is the case, please include the average of all repeated computational experiments, as this would provide smoother loss curves and give readers a clearer understanding of how hyperparameters affect model performance.

Lastly, I am unclear about the authors' answer to my 9th question and the meaning of the term 'Brief Article format'. I would suggest the authors to go through <https://www.nature.com/ncomms/submit/content-types> If they agree with me that they are not constrained by word count, please re-organize the article into sections such as introduction, results and discussion. This will make it easier for readers to fully appreciate their findings.

(Remarks on code availability)

Reviewer #2

(Remarks to the Author)

The authors addressed the comments sufficiently and I have no more major comments.

Minor comment: Extended data figure 6 has title as Ribo-former instead of Ribo-TIE

(Remarks on code availability)

Installation was successful but was having problems with Numpy versions. Might help to add information on Python version required etc.,

Reviewer #3

(Remarks to the Author)

The authors have addressed all of my comments satisfactorily. Consequently, I am entirely satisfied with the current manuscript.

(Remarks on code availability)

Version 2:

Reviewer comments:

Reviewer #1

(Remarks to the Author)

All my previous comments have been addressed in the authors' response, and I have no further concerns.

(Remarks on code availability)

NCOMMS-24-15321-T

Point-by-point response

REVIEWER COMMENTS

Reviewer #1 (Remarks to the Author):

In this study, Clauwaert et al. present RiboTIE, a deep learning-based method to predict novel coding sequences using ribosome profiling data. The RiboTIE pipeline takes the mapped ribosome protected fragments (RPFs) as the input and uses a transformer model to predict the translation initiation sites (TISs) at single-nucleotide resolution. The model was pre-trained on a collection of human ribosome profiling datasets covering different tissues and experimental protocols. Then it was fine-tuned on the target dataset, which requires less computational resources and allows the model to learn dataset-specific rules. The authors demonstrated RiboTIE's performance for the detection of translated ORFs against a few other computational tools. In addition, they conducted ribosome profiling experiments on medulloblastoma cell lines and identified differentially translated non-canonical ORFs (ncORFs) between high/low MYC expression levels. It remains a challenge to decipher the full coding potential of the human genome, and sophisticated tools are needed to improve our understanding of the translational landscape in health and disease. The authors should address the following concerns before publication.

Major comments

1. One of the main concerns I have is regarding the advantages of RiboTIE over TIS transformer, developed by some authors of this work (PMID: 36879896). It seems that both approaches are designed to predict translation initiation sites, use the same transformer architecture called performer and have been trained on the same dataset -- TIS from the Ensembl dataset. On a similar note, in Fig. 2h, the authors used TIS transformer to "confirm" the ncORF predictions by RiboTIE. Does the usage of both tools increase the sensitivity or accuracy of the predictions? The authors should provide guidance on the best strategy to use RiboTIE.

>>>We appreciate the reviewer's comment and we agree that we did not provide sufficient clarification on the distinction in how RiboTIE / TIS transformer function and how they are used for different purposes. We apologize if our use of the word "confirm" led to a lack of clarity and we have now made significant changes in the text to improve this point.

Overall, TIS transformer and RiboTIE both employ transformer models but are different in the data they process and the methodology with which the input vectors are calculated. Perhaps the most clear difference is that RiboTIE uses Ribo-seq data and does not consider nucleotide context, whereas TIS transformer uses only nucleotide context and does not consider ribosome profiling data. At a more

granular level, RiboTIE maps Ribo-seq reads to transcripts without any regard to codon composition / start codon usage, etc. By contrast, TIS transformer models the nucleotide sequence context characteristic of annotated coding sequences (hence biased to AUG start codons) and identifies the probability that all other nucleotides in the transcriptome might act as a start codon. In this way, they can be considered as complementary to each other. Thus, our primary guidance is that these methods can be used as orthogonal approaches to delineate very highly confident ORFs.

Regarding the sensitivity of the approaches – these two methods are independent of each other and thus are not combined in a way that influences the sensitivity of the other.

We have now extended the methods section to include the conditions on selecting ORFs using TIS transformer predictions. Furthermore, we have changed the wording to better reflect our choice of utilizing TIS transformer to further filter down the results—rather than using “confirmed”, as mentioned by the reviewer.

In addition, for researchers interested in working with both RiboTIE and TIS transformer, we have created a new graphic cartoon and re-organized our GitHub page to better guide users on usage of both models.

2. Existing Ensembl annotations may suffer from limitations like under-representation of short and unstable proteins. I wonder how this potential bias affect the prediction of coding sequences. (a) Are the newly predicted translation start sites more enriched in certain transcript biotypes or UTR regions compared to the training dataset? (b) In addition, the authors have proposed some interesting ideas like pretraining the neural network first and then fine-tuned it on separate datasets (e.g., self-supervised and supervised pre-training). How does this affect novel ORF predictions aside from the difference in validation loss?

>>>The reviewer asks an insightful set of questions. To address these individually:

(a) We use Ensembl transcript biotypes. In this approach uORFs, uoORFs, intORFs, dORFs, and doORFs exist on ‘protein coding’ transcripts. Predicted ORFs that do not share a transcript with a canonical coding sequence are either categorized as lncRNA-ORF or CDS variant, where the latter is any ORF sharing the genomic coordinates of its translation initiation or translation termination site with an existing CDS. We do note that some transcript biotypes included in the training set, such as ‘retained_intron’ and ‘nonsense_mediated_decay’ do have a reduced presence for ORFs called over multiple data sets, which may reflect their status as ‘unstable’ or potentially ‘aberrant’ transcripts that are less likely to be translated by ribosomes. To summarize our findings, we have now included **Supplementary Figure 5** (included below) which describes this point on the pancreatic progenitor datasets.

(b) For this question, we have now trained models on the pancreatic progenitor Ribo-seq datasets from scratch. We observe the previously noted improvements in validation loss, ROC/PR AUC performance and training times (see **Supplementary Table 5**).

When evaluating the top predictions for novel ORFs between both models (model score > 0.15), the pretrained models have slightly more calls. This is an expected result because the pretrained models are trained on more data, thus enabling more confidence. Looking specifically into the ncORFs, we see a slight increase in uORFs calls as compared to the other non-canonical ORFs on the transcriptome. We

share these data below, but we decided not to include these data in the revised manuscript as we did not observe any unexpected or biologically relevant insights.

3. Is there a specific reason that RibORF is not included in the benchmarking analyses in Fig. 1b? Apparently, RibORF has been cited and used in this work (e.g., Fig. 2b and 2c).

>>>RibORF was initially not included as the benchmark was performed before evaluation of the Brain samples. We have now included RibORF (v2) as part of the benchmark in **Fig. 1b** in the revised manuscript.

4. The Precision recall (PR AUC) scores reported in Supplementary Table 3 and 4 (0.1-0.2) are significantly lower than the values in Fig. 1b (> 0.5). Are the authors using the same model for these benchmarking experiments? I understand that for each method in Fig. 1b, different ORF subsets were used to evaluate model performance, but how does this filtering step affect RiboTIE's performance?

>>>This is an astute observation which requires a nuanced reply. This manuscript does employ a single RiboTIE model architecture throughout the whole paper, which is detailed in the supplementary files.

The question of why Supplementary Tables 3-4 and Fig. 1b exhibit somewhat different PR AUC scores relates to the different sequencing depths of the different datasets, and thus comparing across datasets is not informative. We are only interested in comparative scores for a given sample dataset.

In essence, RiboTIE processes all positions on all transcripts, and, while ribosome reads are not present along all canonical coding sequences (i.e., not all coding sequences are actively being translated in each sample). But, for our study, we did not alter the target labels of the positive set in accordance to whether a transcript or ORF has a certain number of reads. We decided this approach for the following reasons:

- When no reads are present along a transcript, the transformer model essentially processes a string of zero vectors for that transcript. As such, the presence of a positive target label — or lack thereof — will not negatively affect the model training process (there is nothing to learn from a series of zeros).
- There is no foolproof way to ascertain that a certain number of reads is “to few” to make a prediction. The number of reads that could be present along a transcript or ORF by noise or other factors would be influenced by the transcript/ORF length. Therefore, determining an arbitrary cutoff to alter labels or exclude transcripts from the prediction step is bound to be imperfect.
- We disagree on approaches that seek to subset/generate a custom negative set. These methods generally result in negative sets that don't include the hard-to-predict positions of the transcriptome or apply algorithms that can be easily derived and learned by the neural network. By using the unaltered transcriptome, we ensure that provided evaluation metrics are a good representation of the ability of the tool in the setting it is designed for.
- Altering the prediction set is bound to complicate post-processing or benchmarking attempts.

The inclusion of all transcripts with no alteration of the annotated start sites (even those with no reads) does affect the maximum performance a tool can achieve on a given dataset. Performances in the supplementary files are on the full test set, and therefore lower than performances listed by other tools

that limit the positive/negative set, or those given by the benchmark when the PR/ROC AUC curves of only a select number of sites (as determined by other tools) is evaluated.

To have an idea on how filtering out certain sites along the transcriptome affects the calculated ROC/PR AUC scores, we have included Supplementary Table 6:

Supplementary Table 6: **Comparative performances of RiboTIE based on different subsets of the data.** The model predictions to detect translation initiation sites for each position on the transcriptome can be subsetted as a post-processing step. Using Ensembl translation initiation sites to derive a positive set, the area under the receiver operating characteristic curve (ROC) and area under the precision-recall curve (PR) are calculated. Given is the performance for all positions (total of ~430M), with no conditions for what a valid ORF constitutes, positions that result in an ORF with a valid stop codon on the transcript (stop codon), an ORF length larger than 30 nucleotides (ORF length), and an ATG start codon (ATG start). Additionally, a subset has been selected using a minimum of 20 mapped reads (# Reads) on the transcript as a requirement. The performance and percentage of the total samples when using a combination of all listed conditions (Combined) is listed in the last set of columns. Note that the predictions of RiboTIE are those of the models pre-trained using a supervised learning strategy (see Supplementary Table 5), where the predictions of both models/folds are simply merged to cover the full transcriptome.

dataset	-		Stop codon		ORF length		# Reads		ATG start		Combined		
	ROC	PR	ROC	PR	ROC	PR	ROC	PR	ROC	PR	%	ROC	PR
SRR1802129	0.945	0.020	0.946	0.020	0.943	0.021	0.981	0.041	0.952	0.318	0.6	0.983	0.502
SRR2433794	0.965	0.101	0.966	0.103	0.964	0.103	0.983	0.137	0.966	0.419	1.0	0.981	0.516
SRR2732970	0.969	0.220	0.970	0.222	0.968	0.223	0.986	0.285	0.958	0.399	1.1	0.972	0.487
SRR2733100	0.969	0.215	0.970	0.217	0.968	0.218	0.986	0.279	0.957	0.399	1.1	0.971	0.488
SRR2954800	0.935	0.034	0.935	0.034	0.932	0.036	0.972	0.068	0.942	0.266	0.6	0.974	0.417
SRR8449577	0.958	0.083	0.959	0.084	0.957	0.085	0.983	0.128	0.961	0.382	0.8	0.982	0.514
SRR9113067	0.944	0.016	0.945	0.016	0.942	0.017	0.971	0.024	0.950	0.289	0.9	0.974	0.399
SRR11005875	0.967	0.077	0.968	0.078	0.966	0.080	0.985	0.106	0.969	0.432	1.0	0.984	0.534

5. Supplementary Figure 5 makes little sense to me. The fluctuations in the validation loss are expected due to the stochastic nature of neural network training. But the differences in the model performance seem to be comparable to noise. The authors should use approaches like cross-validation tests to produce replicable results.

>>>The reviewer is correct that the loss curves of **Supplementary Figure 5** are similar, and the reviewer brings up several points that we will individually address.

1. The differences in model performance ‘seem comparable to noise’.

Often, loss curves are represented after smoothing (which we did not do here), as the stochasticity of mini-batch gradient descent results in a (pseudo) random-walk effect. Importantly, this random-walk effect is not a representation of factors of noise in the data and is highlighted given the limited y-axis range of the loss curve.

With our results, we believe that we observe the expected trends of under- and overfitting:

- (a) Models with fewer weights take more training epochs to converge to a minimum validation loss. These curves are typically flatter and oftentimes have a minimum validation loss that is higher than optimal.
- (b) Models with more weights reach a minimum in validation loss more quickly, after which the loss increases (overfitting). This effect is stronger for a higher number of model weights. We interpret this observation as the model having reached a point where additional optimization on

the training samples has an adverse effect on the ability of the model to generalize towards unseen samples.

The set of hyperparameters selected is situated in between the set of hyperparameters that shows signs of underfitting (hyperparameter sets 1,2) and overfitting (hyperparameter sets 7,8), as shown in Supplementary Figure 5. While these observations are subtle, they are in line with the behavior we observed in earlier data when designing this network architecture, for both RiboTIE and TIS transformer, giving us confidence about the set of hyperparameters selected.

2. Use of cross-validation tests

Cross-validation schemes involve unique test sets for each fold, and are therefore not suitable for this setting. The proper way to evaluate whether the differences in model performances are due to noise would be to repeat the training and evaluation phase, which we did many times as part of the overall process of conducting this study.

6. It is unclear how the authors' findings in Fig. 2h and 2j could link back to known or novel cancer biology. I appreciate the authors' effort in conducting new cancer cell line experiments. However, the biological implications of the RiboTIE results are not fully explored. What might be the functional consequences of these differentially translated ncORFs? This work would benefit from a more-in-depth analysis of these findings.

>>>We agree that this aspect of our initial manuscript was not clear enough. Using the set of differential ncORFs in Fig 2h/2j (and listed in Extended Data Table 3), we have revised our manuscript to include the following observations, which supports our ability to connect RiboTIE with biologically-informative results.

1. The ncORFs that are differentially translated in MYC-high vs. MYC-low medulloblastoma cell lines reside on transcripts associated with a neurological location, which reinforces their expected context in a neuronal brain tumor where MYC status influences cell differentiation state. It is not possible to know the biological function of these ncORFs at this time, as this would be outside the scope of the current manuscript. Yet, we feel that their association with neurologically-relevant transcripts recapitulates the known biology regarding the cell-of-origin for MYC-amplified medulloblastomas.
2. The set of uORFs that exhibit widely discordant Spearman correlations between ncORF-CDS pairs (Figure 2i and 2j) are enriched for locations in genes whose proteins are associated with acetylation. We note, however, that the total number of genes used in this analysis is small, limiting our statistical power.
3. Genes with reversed ncORF-CDS correlations between MYC-high and MYC-low are limited to seven (Extended Data Table 4). Although we cannot make broad conclusions, we believe that these are enriched for genes with extensive post-transcriptional control in medulloblastoma. We use 39 medulloblastoma patient samples with matched RNAs and mass spectrometry data (Archer et al, Cancer Cell, 2018). We observe that 4 of the 7 genes highlighted here have

significant protein-RNA dyssynchrony in MYC-high patient tissues compared to MYC-low patient tissues.

We have now included these data as **Extended Data Figure 7** in the revised manuscript.

7. The language used in this paper is too generic. For example, the authors mentioned “Current computational approaches to Ribo-Seq have been developed on a central statistical test using manually curated features...” (p3) and “We have developed RiboTIE as a best-in-class and highly versatile analysis tool ...” (p5). It should be noted that there are numerous computational tools designed for very different tasks in ribosome profiling data analysis. The authors are encouraged to precisely describe the specific task that RiboTIE focused on.

>>> We have revised the manuscript text to improve the language as suggested.

8. I suggest that the authors rewrite the method section to include more details. For example, very little information is given for the training of three types of models (no-pretraining, supervised pre-training and self-supervised pretraining). How are the training datasets constructed in the pretraining and fine-tuning phases? Which model is used to generate results in Fig. 1 and Fig. 2.? In the supplementary material (p3, 1.3.3), the authors mentioned that “multiple models are trained”, so how many models are trained? Also, technical details are missing for Extended Data Figure 7. How are the PCA and t-SNE analysis done?

>>>We apologize that our initial methods section provided insufficient details on several aspects of this work. We have now extended the methods section in the main manuscript to include a subsection on model pre-training (named ‘Model pre-training and fine-tuning’) and added details to the caption referenced here.

9. The authors should consider expanding the main text of this paper. For example, a brief introduction to the existing approaches to delineate ORFs based on ribosome profiling data and mRNA sequence would be beneficial. Additionally, the authors should discuss the potential limitations of their approach.

>>> We appreciate the comment from the reviewer. Because this work is formatted as a Brief Article format, we have been highly compressed for space in our manuscript word count. In the revised manuscript, we have included a short discussion on the major differences, potential drawbacks and future work of the tool near the end of the manuscript. We furthermore added some additional references to some of the other tools (and a supplementary table). We also refer to a recent paper that does a more in-depth description of these tools.

10. Multiple names are used for this work (https://github.com/jdcla/RIBO_former). The authors should use a consistent name for their approach.

>>> This has been addressed and made consistent.

Minor comments

1. In discussion part (p5), “highly versatile analysis tool approach...” seems redundant.
2. Figure 2h, tick labels in y axis should be positive.
3. Figure 2j legend, “(SNAPC5/ACAT1)” I believe it should be “(DEPDC5/ACAT1)”.
4. Supplementary Table 4: “Position” should be “Pre-train”.
5. X axis label is missing for Extended Data Figure 1a.
6. Colormap should be shown for Extended Data Figure 1b.
7. In Extended Data Figure 6, what are colored dots?
8. Extended Data Figure 7, maybe SSH is mistaken for SHH?

>>>We appreciate the reviewer’s close reading of our document and all of the minor comments have been addressed in the revised manuscript.

Reviewer #2 (Remarks to the Author):

The authors present a transformer-based method to identify sites of translation using Ribo-seq data agnostic of start-codon information or ORF characteristics. This is a great approach and a transformer approach is a valuable addition to tools that are currently available for ORF calling. Particularly, powerful as the approach is able to recover ORFs without using the sequence information i.e., without making a restrictive ORF reference library. Although, since there have been several tools published previously, it would be useful to see orthogonal methods of validations of ORFs identified by RiboTIE such as MS, CRISPR-screens etc.,. Lastly, overall, the manuscript is well written but the supplementary methods and figure legends are very brief and need to be expanded.

Major comments

1) While it is not necessary that all the ORFs make stable peptides, it would be important to show if the ORFs identified by RiboTIE are found more often as functional or stable peptides. Independent validations such as using either newly generated or published A) CRISPR-screen data and B) Mass-spectrometry/Immunopeptidomics data to show detection rate of RiboTIE smORFs vs other tools.

>>>The reviewer touches upon a topic of high speculation and much discussion in the non-canonical ORF field. For this revised manuscript, we have conducted the analyses requested by the reviewer – namely, we have compared RiboTIE with ORFquant (the second best tool in our applications) for detection of ncORFs in CRISPR screen data as well as newly-generated mass spectrometry data. These data are included in the manuscript as Supplementary Figure 9 and discussed further below.

However, prior to discussing the data, we feel it is important to point out our concerns with the reviewer’s comment, as we think that the requested analyses are confounded and not clearly meaningful. Our concerns are the following:

1. Regarding CRISPR screening data: CRISPR screening data does not provide ‘independent validation’ or orthogonal evidence for the existence of the ORF, as suggested by the reviewer. CRISPR/Cas9 introduces double-stranded DNA breaks that may suggest ORFs required for cell viability, but may also simply identify genomic regions that cause cell death when cleaved by Cas9, independent of the ORF. Our experience has shown that it is difficult to predict off-target effects for gRNAs to ncORFs, because standardized gRNA prediction outputs such as “on-target scores” used by prediction softwares yield lower, more inconsistent results for ncORFs gRNAs. We have previously tried to parse this with saturation tiling screens of a small number of well-supported ORFs (Hofman et al., Molecular Cell, 2024). We would disagree that CRISPR is useful for ‘independent validations’ of RiboTIE ORFs, as suggested here.
2. Regarding mass spectrometry comparisons: It is not possible to equate the success or failure of a machine learning algorithm for ribosome profiling with the output from mass spectrometry. There are substantial issues with the different technological platforms that negate simple statements such as the one alluded to here, e.g. that the best ORF caller will also ‘more often’ find ORFs with mass spec peptides. There are numerous considerations regarding how the MS is done (how many fractions, enrichments, etc) as well as nuances on calling FDR for peptides. This is a topic that we have previously discussed in a critical perspective piece (Prensner et al., Molecular and Cellular Proteomics, 2023). In many cases, mass spec peptides to ncORFs still represent false positives despite stringent thresholds, which is data that we have accumulated as part of the RiboSeq ORF consortium with PeptideAtlas, and will be presented in a forthcoming manuscript. It is beyond the scope of this paper to establish how often peptides to ORFquant ncORFs compared to RiboTIE ncORFs are false-positive results on manual inspection.

With these considerations in mind, we have done a head-to-head comparison of RiboTIE with ORFquant. We performed two comparisons to address these questions:

1. How often does RiboTIE or ORFquant identify the ncORFs found to be possible medulloblastoma CRISPR hits in our Hofman et al (Mol Cell 2024) paper?
2. How many peptides are found for RiboTIE or ORFquant ncORFs in medulloblastoma tryptic mass spectrometry data?

For (1): We observe essentially equivalent Ribo-seq detection of putative ncORF CRISPR hits by RiboTIE and ORFquant. There are slightly more uoORFs and lncRNA-ORFs found by RiboTIE and a few more uORFs found by ORFquant. However, we would recommend caution in over-interpreting any small differences. Notably, the ncORFs used in that CRISPR screen were largely derived from the GENCODE ncORF set (Mudge et al, Nature Biotechnology, 2022), which itself largely drew from ncORFs found by RiboTaper (the prior version of ORFquant). So the analysis is somewhat circular, finding ORFs with ORFquant that the earlier version (RiboTaper) found in the first place.

For (2): We have now produced high-quality mass spectrometry data for 6 medulloblastoma cell lines, 3 MYC-high (D458, D283, MB002) and 3 MYC-low (ONS76, UW228, DAOY). Each cell line was profiled for tryptic mass spec in 3 replicates with 24 fractions per replicate, for high quality data. 11,090 proteins were detected universally across all samples. To find ncORF peptides, we used FragPipe with a group-

specific FDR of 1%. As seen with other similar studies, there are few ncORF peptides in the tryptic mass spec data. Overall, there are approximately equal numbers of peptides for RiboTIE-nominated ncORFs and ORFquant-nominated ncORFs. However, RiboTIE finds notably more peptides to N-terminal extensions for annotated CDSs compared to ORFquant (discussed below). This is consistent with the optimization of RiboTIE for the detection of translation initiation sites. Significantly more peptides for lncRNA-ORFs are found with ORFquant-defined lncRNA-ORFs. This is likely because ORFquant called a much higher number of lncRNA-ORFs compared to RiboTIE, exemplified in Fig. 1d.

For both approaches (CRISPR and MS), no validated ORFs in the above analyses were shared between both tools.

These data are now included in the revised manuscript as **Supplementary Figure 9**. We have furthermore created **Figure 3** showcasing several examples based on the MS data.

3) According to Fig. 1A the tool also finds peptide extensions of annotated CDS. Are the peptide extensions of annotated CDSs found by RiboTIE also found in MS data, i.e., particularly the sequence of the peptide extended 5' of the annotated start?

>>> As indicated above, we performed mass spectrometry on three MYC-high (D283, D458, MB002) and three MYC-low (ONS76, UW228, DAOY) medulloblastoma cell lines. The data processing was performed using the FragPipe software (<https://fragpipe.nesvilab.org/>), and the Swiss-Prot protein database employed by Fragpipe was enriched independently with the protein sequences derived from ncORFs called by both RiboTIE and ORFquant. Peptides derived from ncORFs were given a lower confidence score (PE=10) in order to exclude peptide matches that also align with proteins listed in Swiss-Prot (PE ∈ [1,5]). In addition, all proteins derived from ncORFs for which resulting peptides were equal to or a substring of any canonical proteins of the Ensembl assembly were filtered out to ensure that matched peptides align to unique findings.

As alluded to above, RiboTIE finds many more N-terminal extensions of annotated CDSs that are supported by mass spectrometry evidence compared to ORFquant, which is consistent with the optimization of RiboTIE for translational start sites. The number of called N-terminal sites by RiboTIE and lncRNA-ORFs by ORFquant is a reflection of the number of calls made by each tool for these ORF types (Fig 1)

We have included these data in the revised manuscript as **Figure 3 and Supplementary Figure 9**.

We have also done a detailed characterization of several N-terminal extension and ncORF peptides for visualization purposes. We show these results here, which highlight the following:

1. An intORF peptide from SCRIB
2. A peptide from an N-terminal extension in the RBMS1 gene
3. A peptide from an uoORF in the ZNF717 gene

2) Similarly to point 1, how evolutionary conserved are the identified smORFs and how often do they have other sequence features such as protein domains as compared to other methods?

>>> To address this question of evolutionary conservation, we applied PhyloCSF (Lin et al.), a multispecies nucleotide sequence alignment tool for determining likelihood of protein-coding regions based on statistical comparison of conservation between known coding regions. To evaluate across multiple tools, we have evaluated the PhyloCSF score on the pancreatic progenitor cells for those ncORFs that were called in all six datasets. Next, to evaluate domain features on ncORFs called by different tools, we did a conserved domain database (CDD) search (<https://doi.org/10.1093/nar/gkac1096>) on the same dataset and evaluated the hit category percentage for each tool according to “Specific”, “Superfamily” and “Non-specific” categories.

In these analyses, RiboTIE scores better than the other tools, alongside Ribo-TISH and RibORF. We have added these analyses to the manuscript as **Supplementary Figure 6**. Furthermore, we have included PhyloCSF and CDD scores to Extended Data Table 5, featuring all ncORFs by all tools when called for all six replicate datasets.

However, we also believe that the biological interpretation of our data is ripe for misrepresentation. In reviewing our data, we are careful from making any conclusions from this analysis towards comparing different ORF callers. So we also add the following context:

1. ncORFs defined by any tool have a mean PhyloCSF score that is overwhelmingly negative (much lower than zero, indicating likely non-coding status). Parsing the biological implications of two very negative PhyloCSF scores is likely not useful, as there is no validated interpretation for such information. That is, a mean score of -3 and a mean score of -5 may be statistically significant, but not biologically meaningful.
2. Both PhyloCSF and CDD-search scores are influenced by post-processing strategies that aggregate predictions per gene. With ribosome profiling reads mapped to the genome and

transcript isoforms sharing genomic regions, it is not possible to detangle which reads were derived from which transcripts (where reads are assigned to all such transcripts). Depending on post-processing strategies, tools might filter out ncORFs calls when they map to genomic regions that also map a canonical protein sequence existent on another transcript isoform. While these strategies are more conservative and likely result in a lower number of false positive predictions, it would reduce overall 'performance' as reflected by PhyloCSF and CDD-search.

Supplementary Figure 6. Evaluation of conservation patterns on ncORFs in pancreatic progenitor cells. (left) PhyloCSF scores evaluate conservation of sequences across organisms. (right) Conserved Domain Database (CDD) searches evaluate the existence of known protein domains in amino acid sequences.

4) How many start isoforms of the known CDS does each tool find? Is the discovery rate higher in general for RiboTIE? How many times did RiboTIE find a TIS for an annotated CDS that was outside of the annotated TIS?

>>> In our resubmission, Extended Data Figure 4 now contains the number of N-terminal extensions (i.e. upstream TIS) and N-terminal truncations (i.e., downstream TIS) predicted by RiboTIE vs. other tools. RiboTIE calls slightly more start isoforms as compared to ORFquant and Rp-Bp, ribotricer and RibORF, while fewer as compared to Ribo-TISH and PRICE.

5) Certain regions of the transcriptome are not mappable considering sequence constraint and the short length of Ribo-seq reads. How do such regions affect the model? Could the authors show how many times annotated ORFs with such unmappable regions are still called by RiboTIE in comparison to other tools?

>>> The reviewer correctly points out the problem with mapping short read lengths to the genome. Based on research by Bekpen et al. (<https://doi.org/10.1093/bfpg/elz016>), we used the six pancreatic progenitor Ribo-seq samples (as in Fig 1b) to evaluate several duplicon gene families within the human genome (NBPF, RGPD, GUSBP, PMS2P, SPATA31, TRIM51, GOLGA8, NPIP, TBC1D3, LRRC37), totalling 692 protein-coding transcripts.

We find that all ORF-calling tools perform poorly for these 692 protein-coding transcripts (see below), which alludes to a problem that is inherent to mapping the reads to the genome, a step performed before running the ORF callers. We do not believe this issue is solvable through predictive tools built on the processed ribosome profiling data. In essence, there is a lack of reads in these regions, as these reads are being discarded during the mapping step, where the maximum number of regions a read can map to is generally put at ~10 (see --outFilterMultimapNmax when using STAR).

Therefore, among these 692 protein-coding transcripts, each tool identifies only a handful of correct ORFs:

- RiboTIE: 14 ORFs
- ORFquant: 23 ORFs

- Rp-Bp: 15 ORFs
- Ribotricer: 6 ORFs
- Ribo-TISH: 0 ORFs
- PRICE: 2 ORFs
- RibORF: 20 ORFs

Given the overall small fraction of correctly called CDSs (e.g., 14 out of 692), it is not possible to determine one tool to be better than another.

6) Could the authors show some examples of some CDSs captured by their method but were not detected by other methods? Plotting a nucleotide resolution coverage view of such examples will help to understand the strengths of RiboTIE specifically in when it is able to detect smORFs that are missed by other tools. Similarly, it would help to show some examples where other tools find a smORF but RiboTIE misses detecting them.

>>> In our revised manuscript, we have added **Supplementary Figure 7** to the main manuscript that displays a random selection of ncORFs (all uORFs here) that were exclusive for RiboTIE (top) or the other tools (bottom). For RiboTIE, examples include ORFs predicted for all six replicates while absent from any replicate by any other tool. For 'other' (bottom), examples include ORFs called by at least two other tools on all six replicates and absent from RiboTIE predictions on any of the six replicate datasets. Given are the ribosome read counts (y-axis) for the replicate pancreatic progenitor cells by positions on the transcript (x-axis). Read counts for the six replicate samples are represented by different colors (but generally overlapping). Areas covering both the predicted uORF (orange) and canonical sequence (gray) are given.

7) On the github page it shows the authors have incorporated a 9-codon search to select alternative ATGs if a non-ATG is identified as a start. It was unclear if this was incorporated in the manuscript - would help to expand on the methods section explaining this. In relation to the same question, how do the authors deal with alternative-start isoforms? Are there multiple isoforms with a different start but having the same splice or same stop codon considered a different ORF?

>>> We have expanded the method section to correctly reflect the 9-codon search explained on GitHub. To clarify, this 9-codon search step is applied for the results on the pancreatic progenitor cells,

medulloblastoma data and adult/fetal brain samples, but not for the construction and calculation of the PR/ROC curves as part of finding the optimal RiboTIE design and benchmark method (basically any PR/ROC AUC score has been derived using the output as is without any post-processing).

Calls on ORFs that share a varying start site on the same transcript are denoted as Extension and truncation (Extended Data Figure 4). If a called ORF shares a genomic start or stop site with an existing CDS existent on another transcript, the called ORF is denoted as a 'CDS variant'.

8) Apart from showing start-codon distribution of RiboTIE identified smORFs, could the authors also show other properties of the smORFs identified such as length distribution, expression and so on.

>>> In line with addressing question 2 of the first reviewer, we have added extra information on the ORFs called for the pancreatic progenitor cells (Supplementary Figure 5).

9) In order to understand what the model is capturing, it would be useful to test and highlight what are the main features the model is capturing such as coverage, length of ORF, read length distribution etc,.

>>> RiboTIE only processes ribosome profiling reads mapped to the transcriptome, and is therefore not able to assess coverage, length of an ORF, etc. directly when making predictions (even though the model might derive these indirectly). Such features are therefore not directly used by RiboTIE. Any other features, such as the evaluation of specific correlations between read counts, their positioning, and read length are expected to be highly nondescript, where it is unfeasible to find overlapping patterns for all data, as the model need to fine-tune on individual datasets to achieve noteworthy performances. While it is likely that the model employs some similar features between datasets, it is out of scope for the manuscript to parse specific model differences across the many datasets employed here.

10) Is one of the main features of the model ORF expression? In that case, is this the reason for low detectability of ncORFs as these are generally found to have lower expression levels compared to uORFs? It would be useful to highlight this as a limitation if that is the case.

>>> The reviewer is right that many research studies have shown lower expression of lncRNAs (and thus lncRNA-ORFs) as compared to protein-translating transcripts, on which uORFs are also present. Importantly, the number of ncORFs are mainly determined by the threshold (0.15) we applied to distinguish our set of positive and negative predictions. We furthermore already established that a higher read depth is the main factor that influences the number of translated ORF calls using RiboTIE.

To look into whether the number of called lncRNA-ORFs by RiboTIE and the number of mapped reads are correlated, we evaluated the reads per base (RPB) of each ORF called by each tool on the pancreatic progenitor cells and categorized them by ORF type. Aligning this with the model outputs for each ORF type reveals the correlation it shares with the RPB, confirming the suggestion of the reviewer.

We see that, indeed, called lncRNA-ORFs have overall lower RPBs as compared to the other subtypes. We furthermore see that the model output of RiboTIE is correlated to the RPB distributions. As such, some tools call more lncRNA-ORFs despite the very low RPB values, and whether these calls reflect false positives is beyond the scope of this paper.

To include more lncRNA-ORFs, it is possible to lower the model output threshold and thereby include a larger set of ncORFs. Currently, the threshold of 0.15 for RiboTIE denotes the most conservative of all tools, as the fewest number of ncORFs are called (while also calling the highest number of canonical CDSs). We are currently looking into optimizing this threshold and implementing post-processing steps that would better distribute the number of calls for each of the ORF categories.

We have included this figure as a **Supplementary Figure 8** and discussed this in the main manuscript.

11) It is unclear for Fig 1b as to how the ROC curve is constructed? Is it that, for each tool's ROC curve, various true sets based on other tools are considered as a positive set? Could the authors draw a Supplementary figure with these ROC curves as well as elaborate in the methods or Supplementary figure legend as to how these were derived to make it easier to understand.

>>> We have extended the main manuscript to include a more detailed description on how the benchmark is performed. In addition, we have added the PR and ROC curves to **Supplementary Figure 4**.

12) In general, the methods section in main/supplementary is quite brief and needs to be expanded.

>>> We have extended the method section as requested by several reviewers.

13) Many of the figure legends do not fully explain the details of the figures. For Fig. 1A it would help to explain briefly for what's in each box in the caption. For Extended data figure 1b, what is meant by coverage in the y axis is unclear. For extended data figure 3a it has not been explained what ec, el, E0 etc., mean? For extended data figure 4, please split into A and B and describe accordingly as these are referenced in different text sections. For extended data figure 6 what do the coloured dots mean?

>>> We have adjusted the figure captions as requested.

14) Could the authors please elaborate on why the TIS transformer model was combined with the RiboTIE approach - what are the limitations of RiboTIE approach that needed both these approaches to be incorporated to find the final set of smORFs for the medulloblastoma dataset? On the intersection of the two methods, why did it lead to a large reduction of numbers?

>>> This question was also asked by Reviewer #1 in comment #1. In summary, both TIS transformer and RiboTIE handle distinct types of data, each with its own hurdles. RiboTIE exclusively uses ribosome profiling sequencing data but not nucleotide sequence contexts. TIS transformer exclusively uses nucleotide sequence contexts but not ribosome profiling data, and is highly biased towards ORFs with an AUG start codon. There is a significant reduction in ncORF numbers mainly because TIS transformer calls very few ncORFs based on nucleotide sequence context alone. The major conclusion is that few ncORFs have a nucleotide sequence context that conforms to annotated CDSs, which is expected since uORFs, dORFs, etc, have distinctly different sequence contexts on an mRNA transcript.

For the question about what limitations in RiboTIE make TIS transformer useful: RiboTIE is limited by the expression profiles of the treated cells, and is bound to be impacted by factors of noise and uncertainty that are linked to the technology. One example of such a factor that would result in a smaller overlap of called translated ORFs between both tools can be explained by how reads mapped to a genomic region covering the introns of multiple transcript isoforms cannot be deconvoluted for Ribo-seq data. These reads will thus be assigned to all transcript isoforms when being processed by RiboTIE. In such cases, TIS transformer can be used in combination with RiboTIE to flag the most likely ORF.

15) In Fig.1b it seems that RiboTISH should have had ~33% positive set as per supp data file but in the figure it looks close to < 10%. Please check and rectify.

16) Also, in the supp file 2: The Rb-bp has positive set 60-90% by calculation but it seems should have been <10%. Please check and rectify.

17) Extended data figure 4 - For the adult brain dataset the numbers don't match the main text. In the figure is it really union or intersect? Please check and rectify.

18) In Fig 2j, the legend says snapc5 gene but in the figure it says depdc5? Are these same genes or is this a typo - please check and rectify.

>>> We thank the reviewer and confirm that these errors have been corrected in the revised manuscript.

Minor comments

19) By the statement “We found that RiboTIE retrieved 64.9% more CDSs (31,431)...”, do the authors mean start-sites? It was unclear whether the authors are comparing start identification or full ORF identification considering only one start-isoform. Please clarify in the text.

>>> We have extended the introduction of the manuscript and method section to further highlight that RiboTIE derives ORF predictions from predicted TISs.

20) The title says detection of sites of RNA translation but the tool does not cover frameshifts or read throughs. It is focused on start-site identification and corresponding ORFs. It would be helpful to be clearer through the title, abstract and introduction that this is a TIS identifier/ ORF caller tool.

>>>We appreciate the reviewer’s comment and we have edited our text carefully to be accurate in our language. Regarding frameshifts and read-throughs, our understanding of the field after implementing numerous ORF callers is that other tools are equally impeded in detecting frameshifts or read throughs as these tools do not include these possibilities when constructing ORF libraries. RiboTIE would allow a post-processing step that could take into account existing or known read throughs and/or frameshifts, but this approach would need more research/time to implement.

21) In Fig 1b, do the authors mean y axis to be the number of ORFs and not “samples”? It might be confusing to readers as is because it sounds as if the plot shows positive set ORFs found for X number of samples.

>>> We have adjusted the manuscript as suggested by the reviewer.

22) The authors use uORF/ncORF interchangeably - would help to be consistent. Especially when referring to ncORF-CDS pairs do the authors mean uORF or is it including lncORF with neighboring protein-coding CDS pairs?

>>> In the manuscript, a ncORF is used for either a uORF, uoORF, intORF, dORF, doORF or lncRNA-ORF. We have included this definition in the manuscript

Reviewer #2 (Remarks on code availability):

The code is well documented on Github and I am able to install it. But there is an error on running the test set given using the current version, so I was unable to evaluate the tool.

>>> We have been able to reproduce the problem of the reviewer and fixed the provided test sets.

Reviewer #3 (Remarks to the Author):

The ribo-seq technique has become indispensable for detecting the translome and its variations, and more particularly for discovering translation events in unconventional ORFs. Several computational methods already exist for the analyses of Ribo-seq data. In this manuscript, the authors create RiboTIE, a method based on transformer models for mapping the global translation of RNA. Their findings indicate that RiboTIE provides higher accuracy and sensitivity in the analysis of ribosome profiling data, compared to other tools. They apply RiboTIE to ribo-seq analyses of normal brain and medulloblastoma cancer samples.

The creative use of the transformer architecture applied to RNA sequences and small ribosome-protected fragments is exactly the sort of cross-field transfer that promises to augment the analytical power needed to tackle the fast-accruing amounts of data generated by recent advances in omics biosciences. We appreciate the extensive description of the algorithm's development methodology including the different schemes attempted for data representation, hyperparameter selection, and pre-training optimization strategies.

This new machine learning-based method will likely set a new standard in ribo-seq analysis.

The following comments are minor:

1. Introduction, second paragraph: the authors allude to certain challenges for computational analyses of Ribo-seq data. However, the provided information is very generic and the non-specialists in this field cannot understand what these challenges are. We therefore recommend specifying these challenges to understand the rationale of this work better.

>>> We have adjusted the manuscript as suggested by the reviewer. We have included more details on the current state of delineation of translated ORFs from ribosome profiling and included references to existing comparative studies.

2. First sentence: "RNA translation is an intricate process that involves the stepwise binding of the 40S and 60S ribosome subunits to RNA, along with multiple eukaryotic initiation factors and other cofactors". This sentence describes part of the translation initiation process, not the actual RNA translation process. Please modify accordingly.

>>> We have adjusted the manuscript as suggested by the reviewer.

3. We appreciate that both ROC AUC and PR AUC were provided as performance indicators. It may perhaps be beneficial to indicate that PR is a more telling measure given the strong imbalance between the classes if the audience is anticipated to comprise many non-experts in machine learning.

>>> We agree with the reviewer and we have adjusted the manuscript as suggested by the reviewer.

4. Fig 2b: why wasn't RibORF included in the benchmarking analyses? Indeed, the authors subsequently (fig 2b,c) compared some features of translated ORFs detected with RiboTIE against those detected with RibORF using the same dataset (reference 14).

>>> We agree with the reviewer's comment and in the revised manuscript, we have included RibORF as part of the main benchmark in Figure 1b.

5. Fig 2d, 2e and 2g: the indicated log₁₀ values appear to be unusually high. Please check.

>>> Provided log values represent reads mapped to the transcriptome, reflecting the counts of mapped reads processed by the model. While this approach omits reads mapping to non-transcribed regions, the number of reads is typically much higher than those mapped to the genome due to the high abundance of transcript isoforms.

6. Supplementary figure 2, SRR1802129, graphs 36 and 37 length: problem with the labels, Y axis.
Supplementary figure 3, SRR8449577, graph 38 length: problem with the labels, Y axis.
Supplementary figure 4, SRR11005875, graphs 36 and 38 length: problem with the labels, Y axis.

>>> We have fixed the figures as requested by the reviewer

7. Data availability: PRJNA1077309 is not available.

>>> We will release the data when the paper is published.

Point by point response to reviewers

NCOMMS-24-15321A

Reviewer #1

I appreciate the authors' revisions to the manuscript, which have addressed most of my concerns. However, I still have a few comments regarding the current form of the manuscript before it can be published.

>>>We are pleased that the reviewer has a favorable view of our work and we did indeed try to address all comments. We apologize that a few of these points were not fully addressed in the first revision of the paper. We have now tried to provide complete answers to all remaining questions.

First, in their response to my fourth question, the authors mentioned: "The question of why Supplementary Tables 3-4 and Fig. 1b exhibit somewhat different PR AUC scores relates to the different sequencing depths of the different datasets, and thus comparing across datasets is not informative."

As a potential user, I kindly disagree with this statement, since I am interested in understanding when and how I can apply a tool like RiboTIE to my dataset. Sequencing depth is likely to be a key determining factor.

>>> We agree with the reviewer and apologize for introducing confusion with our reply. We mentioned:

"The question of why Supplementary Tables 3-4 and Fig. 1b exhibit somewhat different PR AUC scores relates to the different sequencing depths of the different datasets, and thus comparing across datasets is not informative. We are only interested in comparative scores for a given sample dataset. "

With this, we meant that performances between datasets (e.g., SRR1802129 vs. SRR2733100) vary strongly because of sequencing depth, which will have an effect on the performance as we evaluate all annotated CDS during our selection of the optimal architecture of the model. We understand that this does not refer to your question and believe we have caused confusion because of it.

To answer the original question from the first review more specifically:

"Supplementary Table 3 and 4 (0.1-0.2) are significantly lower than the values in Fig. 1b (> 0.5). Are the authors using the same model for these benchmarking experiments?"

Yes, we apply model architecture 4 with input strategy B, as listed in Supplementary Table 3 and 4.

"I understand that for each method in Fig. 1b, different ORF subsets were used to evaluate model performance, but how does this filtering step affect RiboTIE's performance?"

Yes, the exact composition of the positive and negative set does affect the observed performances of the model, as noted by the reviewer. This effect depends on the strategies utilized to derive a positive and negative set.

Different positive and negative sets were utilized throughout the paper to adhere to the context of the research question being answered:

- In the first stage, we looked into optimizing our deep learning implementation for the problem of parsing ribosome profiling data for detecting translated ORFs (hyperparameter tuning, input

token strategy, pre-training). Performances are evaluated on the full dataset as we are convinced this is the most unbiased approach (**see first point-by-point reply**).

- In the second stage, we sought to evaluate our model against previous studies. With previous studies all handling custom ORF libraries derived from the original data, benchmarking is not possible in the traditional, unbiased way. Our answer to this problem was to evaluate RiboTIE one-on-one with each of the listed tools, as RiboTIE provides predictions along the full transcriptome. In this stage, the positive and negative sets are determined by the other tools. We note that performances between tools, because of their different ORF-library construction strategies, should not be compared. We have added an extra note on this in the manuscript.
- In the third stage, we look at the positive predictions of each tool together and evaluate their properties. The strategy to construct a positive set is described in the Methods section.

To address the reviewer's question regarding the differences in performances in Fig 1b (benchmarked performances), we do not deem it in-scope of the manuscript to look into each of the strategies applied by other tools to derive their ORF libraries. The goal of the benchmark is to show RiboTIE to deliver superior performances against each of these tools.

To answer the reviewer's question, we have included **Supplementary Table 6** to give the reader insight into how differing filtering approaches affect the performances for different datasets.

As a potential user, I kindly disagree with this statement, since I am interested in understanding when and how I can apply a tool like RiboTIE to my dataset. Sequencing depth is likely to be a key determining factor.

>>>We understand the importance of understanding when and how RiboTIE can be applied. We find that RiboTIE performs better than previous methods (Fig 1b, Extended Data Table 2) on a variety of datasets covering a wide range of sequencing depths (Extended Data Table 1). We believe this fact sufficiently addresses the question of when to apply RiboTIE – in essence, we have applied RiboTIE on data from numerous public datasets generated by differing lab protocols for RiboSeq and widely different sequencing depths. As with any machine learning approach, deeper sequencing with high-quality libraries of diverse RNA molecules (e.g. not simply sequencing more PCR duplicates) will likely optimize the results obtained, but it is not possible to provide a single, one-size-fits-all “best practices” for sequencing depth as this will depend on the scientific question being pursued.

Thus, as is the case with any piece of software, the question on how to apply RiboTIE depends largely on the user and their research objective. On the effect of sequencing depths on RiboTIE, we have found the tool to return relatively stable accuracies for any given threshold, independent of sequencing depth. This is best shown on the large number of model outputs we obtained for the adult brain samples (see Extended Data Table 1 for sequencing depths). Evaluating the model output distributions for the top 100,000 ranked predictions, we see strong differences between number of predictions above a certain threshold:

However, when evaluating the distribution of ORF types of the positive set, we see the number of called annotated CDS to be relatively stable (reflecting the model accuracy).

These figures, ranked by number of predictions above the threshold, and in-line with sequencing depths reported (Extended Data Table 1), show that sequencing depth has an effect on the number of positive predictions returned by RiboTIE rather than the composition of the predicted set. As such, the default behavior for the user is to have a stable accuracy, with varying number of positive results, which we deem to be the expected behavior of a tool. Therefore, we do not think the user needs to alter the input parameters of RiboTIE based on the sequencing depth of the input data.

To better reflect the behavior of the model output, the figure was generated directly from the model outputs without any additional post-filtering steps (e.g. start codon, etc), as were performed for the results in the paper (see **Methods**). We have added this figure in the supplementary materials (**Supplementary Figure 16**).

Lastly, adjustments of the threshold to determine the positive set and various filtering behaviors can be set from the RiboTIE command line. It is up to the user to determine these alternate filtering steps in accordance to their objectives, e.g. by trading in higher precision for recall or vice versa. These can be easily verified by looking at the number of predicted annotated CDSs in the result table.

We hope we have more specifically answered the reviewer's concerns.

Second, the authors' response to my fifth question is not fully convincing. I have some expertise in the field of deep learning, and I still do not see why architecture 4 is the optimal. Furthermore, the authors stated: "The proper way to evaluate whether the differences in model performances are due to noise would be to repeat the training and evaluation phase, which we did many times as part of the overall process of conducting this study." If that is the case, please include the average of all repeated computational experiments, as this would provide smoother loss curves and give readers a clearer understanding of how hyperparameters affect model performance.

>>>To remind the reviewer, the prior fifth question relates to the original Supplementary Figure 5, which is Supplementary Figure 11 in the revised manuscript last submitted. This figure shows loss curves for the model architectures.

We first address the first part of the question:

Second, the authors' response to my fifth question is not fully convincing. I have some expertise in the field of deep learning, and I still do not see why architecture 4 is the optimal.

Our selection of architecture 4 based on the hyperparameter strategy adopted in the manuscript is twofold:

- In accordance with best practices and convention in the field, hyperparameters are selected through a classic train/test/validation set split. Architecture 4 is selected simply because it returned the lowest validation loss (Supplementary table 4). The value for Architecture 4 in Supplementary Table 4 is 1.095×10^{-3} , but we agree that values for other architectures are similar, eg 1.099×10^{-3} for Architecture 5. This indicates that several architectures mainly reproduce each other, which is a good thing that indicates the suitability of analyzing ribosome profiling data with machine learning. This brings us to the second point below.
- The strongest difference between the models is situated in training time and virtual memory required by the GPU. The number of epochs required for models with fewer parameters was less stable, sometimes only converging after 20+ epochs, where the virtual memory required for larger models put possible hardware constraints on the tool. Architecture 4 balances these two factors.

Furthermore, the authors stated: "The proper way to evaluate whether the differences in model performances are due to noise would be to repeat the training and evaluation phase, which we did many times as part of the overall process of conducting this study." If that is the case, please include the average of all repeated computational experiments, as this would provide smoother loss curves and give readers a clearer understanding of how hyperparameters affect model performance.

Unfortunately, we do not have access to these preliminary data. This project was initiated at Ghent University (Belgium) and completed at the University of Michigan (USA). As part of that transfer, older data was not preserved if we deemed it to be minimally informative for the implementation of RiboTIE, and the final data listed in the manuscript was only run once after finalization of the experimental set-up. We apologize for this inconvenience but we trust that the reviewer appreciates this situation.

Yet, we recognize the reviewer's comment. Thus, to address this request, we have re-run the model using the RiboTIE version the data was initially trained on (v0.3.3). For completeness of the details, we were unable to run this on the exact older PyTorch version due to software problems related to updates in PyTorch which complicate the usage of the older version at the present time.

We have rerun the training process three times for five of the architectures (Architecture 1,3,4,5,8), including those with the fewest and most weights. Training stops after a model does not reduce its validation score after 6 epochs (early stopping), resulting in total training times that vary between models.

The data look similar to those generated by the validation loss curves in the manuscript, with similar trends as the ones described in the first reply: models with fewer weights (Architecture 1) converge more slowly while models with more weights (Architecture 8) have a faster upwards trend of the validation loss.

To address the concern the reviewer raised in the second review that Architecture 4 is not necessarily the model with the lowest validation loss when repeating the experiment, we agree with this point. This is reflected by the repeated experiments where both the model with the fewest and most weights in this experiment would have been selected (Architecture 1, 8). In essence, several Architectures appear equivalent with some variability between each experimental analysis, which suggests that there is no significant difference in their performances in terms of validation loss. **This is concordant with the results in our paper.** As mentioned above, however, Architecture 4 was selected not only based on the validation loss in our initial experiment, but also based on factors such as the amount of computational resources needed to run these models.

Moreover, we have spent time deeply considering these results. After further consideration of what causes fluctuations in the validation loss between epochs to be larger than expected (when compared to the difference of minimum validation loss achieved by the models), we believe two factors to be of interest, if researching this more closely:

- There is an extreme imbalance of the positive and negative set. It is possible this contributes to the observed effect.
- The approximation algorithm used by the PERFORMER network. PERFORMERS use the FAVOR+ (Fast Attention Via Positive Orthogonal Random Features) approximation algorithm, which affects the prediction output of the model. It is possible fluctuations are (partially) caused by this setting.

It is also of interest to check whether both factors have a compounding effect.

We do not think further research into this behavior is in-scope for the manuscript. The PERFORMER architecture was essential in allowing transformer networks to be used on 30,000+ sequence inputs at the time. Figuring out whether this factors into the observed effect would be beyond the scope of this manuscript.

Crucially, models with similar validation loss performances are expected to perform similarly. As the models have similar validation losses that are reproduced on the replicated data, we are confident that

selection of one over the other will be irrelevant to the conclusions of the manuscript, especially given the wide differences between the performance of RiboTIE and previous tools.

In summary, we agree with the reviewer that it is unclear which architecture is optimal based on the data of the validation loss curves. We argue multiple Architectures could have been used without affecting the results of the paper. Nonetheless, we believe the approach used in the manuscript is consistent with good practices in the field of machine learning, using the data and information available to us at the time of construction of the model, where the initial data supported Architecture 4. We also point out that we endeavored to balance validation loss with the intensity of the computational needs to run the model for users. To highlight this point, we emphasize the fact that Architecture 4 has ideal training times (training time converges after a few epochs and is 45 minutes as compared to 1h30min for architecture 8) and requires less than half of the virtual memory of architecture 8 (alleviating hardware constraints). In comparison to Architecture 1, Architecture 4 is likely to benefit from the added model parameters taking into account the model pre-training approach that was successfully explored at a later step in the manuscript. In addition, models with fewer parameters have unstable training times, given that convergence sometimes required 20+ epochs.

Overall: while re-running the Architectures retrospectively in a slightly different context (e.g. PyTorch versions, etc) provides a slightly different result, this variance is in-line with our expectations and does not impact the strength or robustness of our findings in this manuscript, nor will it appreciably impact users of RiboTIE.

Rebuttal-only Figure: Validation loss of different model architectures as computed after each epoch. Each set of architectures was re-trained three times from scratch.

We did not include these data as part of the manuscript for two important reasons:

- It does not seem advisable to merge the data with our previous results. Data was trained on different GPUs, resulting in different batch sizes, which not only affects the number of training steps between each epoch, but is also expected to affect the training behavior to some extent. There is also the difference in PyTorch versions, the effect of which is unknown.
- We believe these new results would obfuscate the message to the reader rather than inform. The new results would not as clearly reflect the thought-process taken that were based on the original results. While we appreciate the questions raised by the reviewer, we do not think that the reader would benefit from these additional experiments.

Lastly, I am unclear about the authors' answer to my 9th question and the meaning of the term 'Brief Article format'. I would suggest the authors to go through <https://www.nature.com/ncomms/submit/content-types> if they agree with me that they are not

constrained by word count, please re-organize the article into sections such as introduction, results and discussion. This will make it easier for readers to fully appreciate their findings.

>>>We have re-formatted the article in accordance with the suggestions of the reviewer.

Reviewer #2 (Remarks on code availability):

Installation was successful but was having problems with Numpy versions. Might help to add information on Python version required etc,.

>>>We have added a minimum python requirement on the GitHub page. Otherwise, we have reinstalled RiboTIE in a clean environment without issues. We recommend installing the tool in a new environment without any pre-installed packages.